# Systematic analyses of the *MIR172* family members of *Arabidopsis* define their distinct roles in regulation of *APETALA2* during floral transition

**Diarmuid S. Ó'Maoiléidigh**[1,2], **Annabel D. van Driel**[1], **Anamika Singh**[1], **Qing Sang**[1], **Nolwenn Le Bec**[1], **Coral Vincent**[1], **Enric Bertran Garcia de Olalla**[1], **Alice Vayssières**[1], **Maida Romera Branchat**[1], **Edouard Severing**[1], **Rafael Martinez Gallegos**[1], **George Coupland**[1]*

**1** Department of Plant Developmental Biology, Max Planck Institute for Plant Breeding Research, Cologne, Germany, **2** Institute of Systems, Molecular, and Integrative Biology, University of Liverpool, United Kingdom

☺ These authors contributed equally to this work.
* coupland@mpipz.mpg.de

**Citation:** Ó'Maoiléidigh DS, van Driel AD, Singh A, Sang Q, Le Bec N, Vincent C, et al. (2021) Systematic analyses of the *MIR172* family members of *Arabidopsis* define their distinct roles in regulation of *APETALA2* during floral transition. PLoS Biol 19(2): e3001043. https://doi.org/10.1371/journal.pbio.3001043

**Data Availability Statement:** The next generation sequencing data of the genomic DNA from the mir172 quintuple mutant is available from the

## Abstract

MicroRNAs (miRNAs) play important roles in regulating flowering and reproduction of angiosperms. Mature miRNAs are encoded by multiple *MIRNA* genes that can differ in their spatiotemporal activities and their contributions to gene regulatory networks, but the functions of individual *MIRNA* genes are poorly defined. We functionally analyzed the activity of all 5 *Arabidopsis thaliana MIR172* genes, which encode miR172 and promote the floral transition by inhibiting the accumulation of APETALA2 (AP2) and APETALA2-LIKE (AP2-LIKE) transcription factors (TFs). Through genome editing and detailed confocal microscopy, we show that the activity of miR172 at the shoot apex is encoded by 3 *MIR172* genes, is critical for floral transition of the shoot meristem under noninductive photoperiods, and reduces accumulation of AP2 and TARGET OF EAT2 (TOE2), an AP2-LIKE TF, at the shoot meristem. Utilizing the genetic resources generated here, we show that the promotion of flowering by miR172 is enhanced by the MADS-domain TF FRUITFULL, which may facilitate long-term silencing of *AP2-LIKE* transcription, and that their activities are partially coordinated by the TF SQUAMOSA PROMOTER-BINDING-LIKE PROTEIN 15. Thus, we present a genetic framework for the depletion of AP2 and AP2-LIKE TFs at the shoot apex during floral transition and demonstrate that this plays a central role in floral induction.

## Highlights

1. Characterization of clustered regularly interspaced short palindromic repeats (CRISPR)-induced mutant alleles and fluorescent protein reporter constructs of entire *MIR172* family.

BioProject NCBI database (http://www.ncbi.nlm.
nih.gov/bioproject/646473). SubmissionID:
SUB7773938, BioProject ID: PRJNA646473 The
RNA-Seq time-courses comparing the
transcriptomes of apices between Col-0 and
mir172abd is available from the BioProject NCBI
database. BioProject ID: PRJNA669254 Data
underlying Figs and S Figs are available in S1 Data
and S2 Data.

**Funding:** D.S.Ó'M was supported by a von
Humboldt Foundation post-doctoral fellowship
(https://www.humboldt-foundation.de/). SQ was
funded by a studentship from the China
Scholarship Council (https://www.
chinesescholarshipcouncil.com/). AS was funded
by a DAAD funded exchange studentship (https://
www.daad.de). GC was funded by a grant from the
Deutsche forschungsgemeinschaft (https://www.
dfg.de/, CO 318/11-1), a grant from the ERC
(https://erc.europa.eu/, N°339113 – HyLife) and is a
member of a DFG-funded Cluster of Excellence
(https://www.dfg.de/, EXC 2048/1 Project ID:
390686111). The laboratory of GC receives a core
grant from the Max Planck Society. The funders
had no role in study design, data collection and
analysis, decision to publish, or preparation of the
manuscript.

**Competing interests:** The authors have declared
that no competing interests exist.

**Abbreviations:** ANOVA, analysis of variance; AP1,
APETALA1; AP2, APETALA2; AP2-LIKE,
APETALA2-LIKE; Cas-9, CRISPR associated
protein-9; CLN, cauline leaf number; CRISPR,
clustered regularly interspaced short palindromic
repeats; DCL1, DICER-LIKE 1; DTB, days to bolting;
DTF, days to the first flower opening; FPKM,
Fragments Per Kilobase of transcript per Million;
FT, FLOWERING LOCUS T; FUL, FRUITFULL; GUS,
β-glucuronidase enzyme; LD, long-day; MIR,
MICRORNA; miRNA, microRNA; NVG, NLS-Venus-
GUS; PIPE, Polymerase Incomplete Primer
Extension; PP2A, PROTEIN PHOSPHATASE 2A;
PPT, phosphinothricin; RNA-seq, RNA-sequencing;
RT-PCR, reverse transcription PCR; RT-qPCR,
quantitative reverse transcription PCR; SAM, shoot
apical meristem; SEM, standard error of the mean;
SD, short-day; sgRNA, single guide RNA; SMZ,
SCHLAFMÜTZE; SNZ, SCHNARCHZAPFEN; SOC1,
SUPPRESSOR OF OVEREXPRESSION OF
CONSTANS1; SPL, SQUAMOSA PROMOTER
BINDING PROTEIN-LIKE; TAIR, The *Arabidopsis*
Information Resource; TF, transcription factor;
TLN, total leaf number; TOE1, TARGET OF EAT1;
TSF, TWIN SISTER OF FT; WT, wild-type.

2. microRNA172 activity at the shoot apical meristem (SAM) is crucially important for floral transition under short-day (SD) conditions.

3. Coordinated transcriptional and posttranscriptional mechanisms repress the activity of APETALA2-LIKE (AP2-LIKE) transcription factors (TFs) during flowering.

## Introduction

Transcription factors (TFs) and microRNAs (miRNAs) are well-studied regulators of development in animals and plants. TFs regulate the transcription of their targets by interacting with *cis*-regulatory elements in the genome, whereas miRNAs bind to mRNAs to induce transcript degradation or to inhibit their translation. The mRNAs of TFs are prominent targets of miRNAs, and in animal systems feedback loops between miRNAs and their target, TFs have been shown to confer robustness to gene expression patterns during development [1]. Plant and animal miRNAs are similar in length (20 to 24 nucleotides); however, sequence complementarity between plant miRNAs and their target mRNAs is more extensive than in animals, which is consistent with the higher proportion of miRNA-targeted genes in animals than in plants [2]. The increased specificity of plant miRNAs is offset by the expansion of gene families during plant evolution, particularly TFs, with most plant miRNAs controlling the activity of multigene TF families [3,4].

Plant miRNAs are encoded by families of *MICRORNA* (*MIR*) genes. The roles of individual MIRs are relatively poorly understood because loss-of-function mutant alleles are not available for most of them, and functional redundancy is assumed to occur among the members of a single *MIR* family. To understand the interactions between miRNAs and their targets, most research has been limited to the use of miRNA mimicry strategies or the expression of miRNA-resistant versions of the target genes [5–10]. Both of these approaches have been useful but are subject to artefacts caused by off-target effects or ectopic expression, and they do not offer insights into the contribution of individual *MIR* family members to developmental regulatory networks. The advent of clustered regularly interspaced short palindromic repeats (CRISPR)-CRISPR associated protein-9 (Cas-9) systems to induce mutations at specific genomic loci provides a new opportunity to study miRNA functions in plants [11–14]. *MIR*s can be inactivated by different types of mutation that could be induced by CRISPR, such as deletion of the coding sequences of the miRNA/miRNA* duplex or mutagenesis of the sites required for DICER-LIKE1 processing of the pri-miRNA [15,16]. Here, we follow both strategies to inactivate all 5 members of the *MIR172* gene family in *Arabidopsis thaliana*.

The targets of miR172 are the mRNAs of the APETALA2-LIKE (AP2-LIKE) TF family, and this interaction is conserved across spermatophytes [17,18]. In *A. thaliana*, there are 6 members of the AP2-LIKE TF family APETALA2 (AP2), TARGET OF EAT1 (TOE1), TOE2, TOE3, SCHNARCHZAPFEN (SNZ), and SCHLAFMÜTZE (SMZ). The founding member, AP2, was initially characterized because *ap2* mutations exhibit floral organ defects that led to it being assigned A-class function in the ABC model of flower development [19,20]. Subsequently, *AP2* and the other *AP2-like* genes were shown to act as repressors of the floral transition [7,8,21–23]. Ectopic expression of *MIR172* phenocopies strong *ap2* mutants, whereas ectopic expression of a miR172-resistant version of *AP2* mRNA (*rAP2*) or *TOE3* (*rTOE3*) from the viral *35S* promoter produced strongly indeterminate flowers [8,23,24], which is partially due to the negative regulation of the C class gene *AGAMOUS* by these AP2-LIKE TFs

[21,25,26]. Ectopic expression of *AP2* and *rAP2* also dramatically delays flowering [8], which was also observed when miR172 susceptible and resistant versions of *TOE1* and *SMZ* were ectopically expressed [7,27]. In contrast, plants lacking the activities of all *AP2-LIKE* genes flower extremely early, producing only a few leaves prior to flowering [21].

Genomic analyses of AP2 and SMZ binding sites revealed that they have independent and overlapping targets during development. For example, both AP2 and SMZ bind to regulatory regions of the floral meristem identity gene *APETALA1* (*AP1*) and the floral integrator gene *SUPPRESSOR OF OVEREXPRESSION OF CONSTANS1* (*SOC1*), with plants harboring combinations of mutations in *AP2-LIKE* genes containing higher levels of *AP1* and *SOC1* mRNAs relative to wild-type (WT) plants [7,21]. In contrast, SMZ, but not AP2, directly negatively regulates the florigen-encoding gene *FLOWERING LOCUS T* (*FT*) [7,21], which is transcribed in the vasculature of leaves, while its protein product moves to the shoot apical meristem (SAM) to induce flowering [7,28–30]. TOE1 also directly negatively regulates *FT* expression, although the regulatory sites it binds to are different to those of SMZ [7,27]. Whether the separation of AP2 and SMZ functions as well as those of other AP2-LIKE TFs is based on biochemical activity or expression pattern differences is currently unclear.

Much of our knowledge of how miR172 interacts with the *AP2-LIKE* genes is derived from transgenic plants exhibiting ectopic expression of a *MIR172* gene or a miR172-resistant version of an *AP2-LIKE* gene [7,8,22–24,27,31]. These approaches indicate that ectopic expression of an *rAP2-LIKE* gene results in a more severe phenotype when compared to ectopic expression of the *miR172*-sensitive version of the same gene [7,8,24,31]. Three isoforms of miR172 are present in *A. thaliana* and are encoded by *MIR172A/B*, *MIR172C/D*, and *MIR172E* [8]. Ectopic expression of any of the *MIR172* genes results in early flowering [22,23,27,32], whereas ectopic expression of a *MIM172* mimicry construct delays flowering [5]. Mutant alleles for *MIR172A* and *MIR172D* have been identified, although these alleles do not appear to be null mutants and neither of them causes late flowering [6,33]. The *mir172a-1* allele causes abaxial trichomes to form earlier than corresponding controls, whereas *mir172d-1* is defective in floral meristem determinacy [6,34]. Although a mutant allele for *MIR172B* is not available, this gene has been identified as a direct target of several members of the SQUAMOSA PROMOTER BINDING PROTEIN-LIKE (SPL) TF family [6,35] that contribute to juvenile to adult vegetative phase change and the age-dependent pathway to flowering [6,9].

The mRNAs of the *SPL* gene family are targeted by miR156 and miR157, and these families are evolutionarily conserved across all embryophytes [17,36,37]. This TF-miRNA module regulates diverse developmental processes including trichome distribution, plant architecture, and flowering time [17,36]. In *A. thaliana*, the protein sequences and structures of the 17 SPL TF family members vary [38], suggesting that there is biochemical diversity among SPL TFs that is further pronounced by their diverse patterns of gene expression [39]. In many flowering plants, miR156 levels decrease with plant age with *SPL* gene expression being inversely correlated [9,40–44]. Ectopic expression of *MIR156* coding genes delayed flowering [40,42–44], while generating *miR156*-resistant versions of *SPL* transcripts [6,9,41,44] or depleting miR156 through the expression of a *MIM156* mimicry construct [9,42,43] promoted flowering.

In *A. thaliana*, the effect of perturbing *SPL* or *MIR156* gene function on flowering time is much more pronounced under noninductive photoperiods when compared to inductive photoperiods [35,45]. Recently, *spl15* single mutants were shown to flower extremely late under short-day (SD) conditions, whereas *spl15* plants grown under long-day (LD) conditions flowered only slightly later than WT [35]. SPL15 appears to act independently of *FT* and its paralog *TWIN SISTER OF FT* (*TSF*), which can bypass the requirement for *SPL15* [35,41]. Functional analysis of SPL15 activity revealed that it bound to regulatory regions near *MIR172B* and the meristem identity gene *FRUITFULL* (*FUL*) to promote their expression [35], as observed for

other members of the SPL family [9,46]. *FUL* encodes a MADS-domain containing TF that promotes the floral transition, fruit development, and floral meristem identity and was implicated in the control of inflorescence meristem determinacy, a process termed "global proliferative arrest" [47–51]. Recently, FUL was shown to directly negatively regulate transcription of several *AP2-LIKE* genes such as *SNZ*, *TOE1*, and *AP2* during inflorescence development [47], although it is unclear if the regulation of these genes by FUL is important during the floral transition.

We present a comprehensive analysis of the role of *MIR172* genes in floral transition by generating novel gene expression reporters and CRISPR-Cas-9-induced mutations for each of the 5 *MIR172* members. Detailed phenotypic analysis of single and higher-order mutants as well as confocal microscopy determine the contribution of individual *MIR172* genes to floral transition and demonstrate the role of miR172 is most critical under SDs. Further genetic and imaging experiments show that one function of miR172 at the shoot apex is to reduce AP2 protein levels during floral transition. By generating combinations of *miR172* mutations, *ful* and *spl15*, we show that miR172 and FUL act in parallel to promote floral transition, and their activities are important downstream of SPL15. We propose that AP2 and FUL act at the core of a bistable switch mechanism regulating early stages of the floral transition at the SAM.

## Results

### Expression analysis of the *MIR172* family in long days

The expression patterns of the *MIR172* genes were previously assessed by semiquantitative reverse transcription PCR (RT-PCR) in whole seedlings [22]; however, the expression of these genes in the SAM was not determined. Therefore, we harvested RNA from apical tissue of LD-grown plants through a developmental time course and performed quantitative reverse transcription PCR (RT-qPCR; Fig 1A–1F). We analyzed the accumulation of the *MIR172* precursor RNAs, because the mature forms encoded by each precursor are too similar to distinguish by this method [8]. Under these conditions, the floral transition occurs between 14 and 17 days after germination, as defined by the level of *AP1* transcript (Fig 1A) and microscopic examination (see later). *MIR172A* was more highly expressed at 7 days, was at trough levels at 10 days, and then increased again through 12, 14, and 17 days (Fig 1B). In contrast, *MIR172B* and *MIR172D* RNAs were at trough levels during vegetative development, but increased during floral transition (Fig 1C–1E). Thus, the abundance of these 3 *MIR172* RNAs increased most sharply at the apex during the floral transition and was closely correlated with *AP1* expression, although significant increases in *MIR172A* RNA were observed prior to *AP1* expression (Fig 1B). *MIR172C* and *MIR172E* were very lowly expressed in samples derived from vegetative plants or plants undergoing the floral transition; however, *MIR172C* was expressed at 21 days (Fig 1D–1F).

To improve the resolution of the spatial expression patterns, we cloned the 5 intergenic regions containing the *MIR172A-E* genes and replaced the miR172 hairpin regions with the coding region for a nuclear localized fusion between the fluorescent protein Venus and β-glucuronidase (NLS-Venus-GUS, NVG) (S1 Fig). Notably, the size and genomic structure of each of the *MIR172* genes varied greatly (S1 Fig), which might contribute to their diverse expression patterns. We analyzed several independent transformants of each construct and selected at least 2 for more detailed analysis (Fig 1, S2 and S3 Figs). Apical tissue of these lines was harvested at the same time points described for the RT-qPCR experiments and analyzed by confocal microscopy (Fig 1, S2 and S3 Figs). Only *MIR172A-NVG*, *MIR172B-NVG*, and *MIR172D-NVG* were reproducibly expressed at the shoot apex with each of their temporal expression patterns largely in agreement with the RNA profiling study (Fig 1G–1I, S2A–S2C

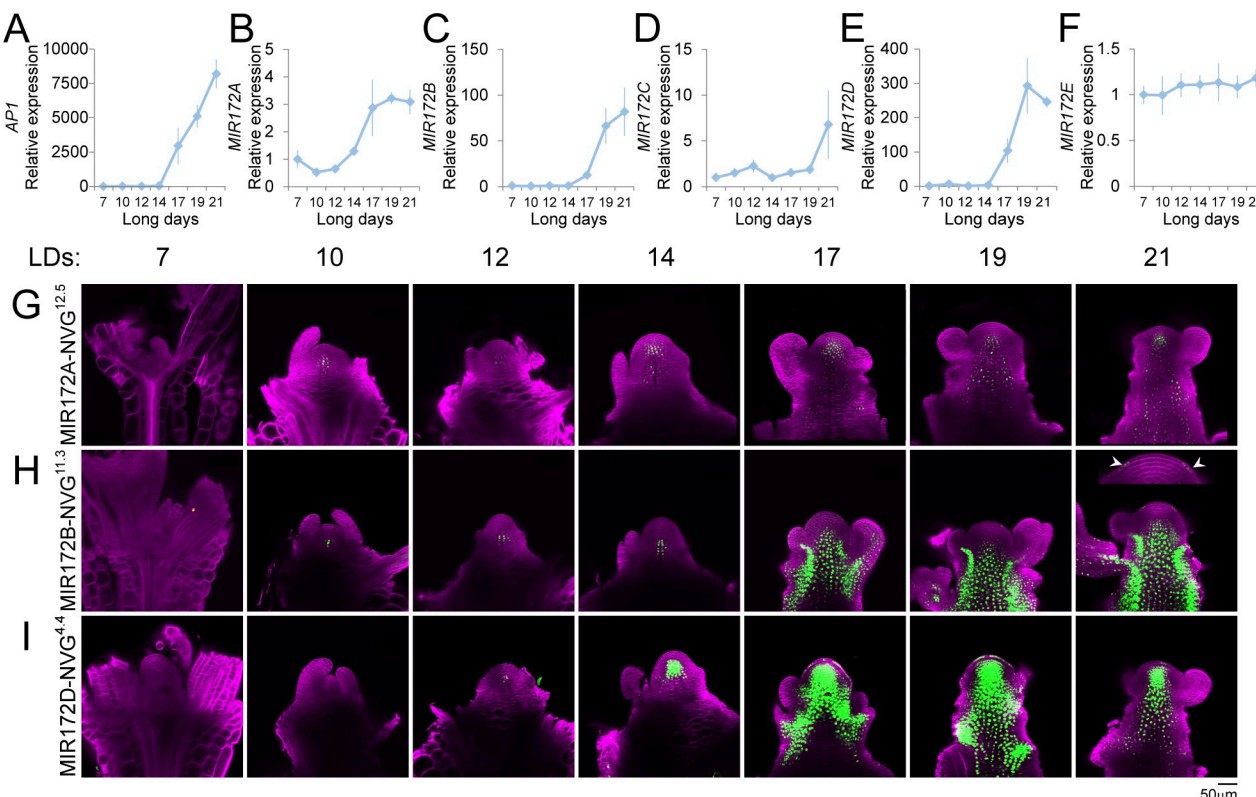

**Fig 1. Expression patterns of the *MIR172* gene family in the shoot apex.** (A–F) Graphs depicting RT-qPCR results of (A) *AP1*, (B) *MIR172A*, (C) *MIR172B*, (D) *MIR172C*, (E) *MIR172D*, and (F) *MIR172E* expression in materials derived from the shoot apex of LD-grown plants that were harvested at the indicated times after germination. Expression was normalized to 7d samples. Error bars indicate SEM of 3 independent biological replicates. Data underlying panels A–F are provided in S1 Data. (G, H) Confocal laser scanning micrographs of the shoot apices of (G) MIR172A-NVG, (H) MIR172B-NVG, and (I) MIR172D-NVG transgenic plants grown in LD conditions and harvested at the indicated times after germination. (H) Note fluorescence in L1 at 21d *MIR172B-NVG*[11.3] meristem (inset, arrowheads). Fluorescence from the Venus protein is artificially colored in green, and the fluorescence from the Renaissance dye is artificially colored in magenta. LD, long-day; NVG, NLS-Venus-GUS; RT-qPCR, quantitative reverse transcription PCR; SEM, standard error of the mean.

and S2F Fig). Expression of *MIR172A-NVG* and *MIR172D-NVG* was detected from around day 14 in and below the L3 layer of the SAM (Fig 1G–1I, S2A–S2F Fig), whereas *MIR172B-NVG* expression was mainly detected in the rib meristem and the vasculature extending from the main shoot to floral buds (Fig 1H, S2B and S2C Fig). Fluorescence was also observed in the L1 of *MIR172B-NVG* (Fig 1H–21d, S2B and S2C and S3D Figs) and *MIR172D-NVG* (Fig 1I—17–19d, S2F Fig) transgenic plants, although this occurred only at specific times. *MIR172C-NVG* was mainly expressed in the vasculature of the inflorescence stem (S2D and S2E and S3G–S3I Figs); however, we were unable to detect expression of *MIR172E-NVG* in the shoot apex. Taken together, these data demonstrate that the *MIR172* genes exhibit diverse temporal and spatial expression patterns and that *MIR172A*, *MIR172B*, and *MIR172D* increase in expression in or near the shoot apex before and during the floral transition, whereas *MIR172C* is mainly expressed in the inflorescence stem, and *MIR172E* activity could not be detected in the apex of LD-grown plants.

## Inactivation of all *MIR172* genes using genome editing

Null or strong loss-of-function mutants for the *MIR172* genes are not available in *A. thaliana*, although alleles that reduce the activity of *MIR172A* and *MIR172D* have been described [6,34].

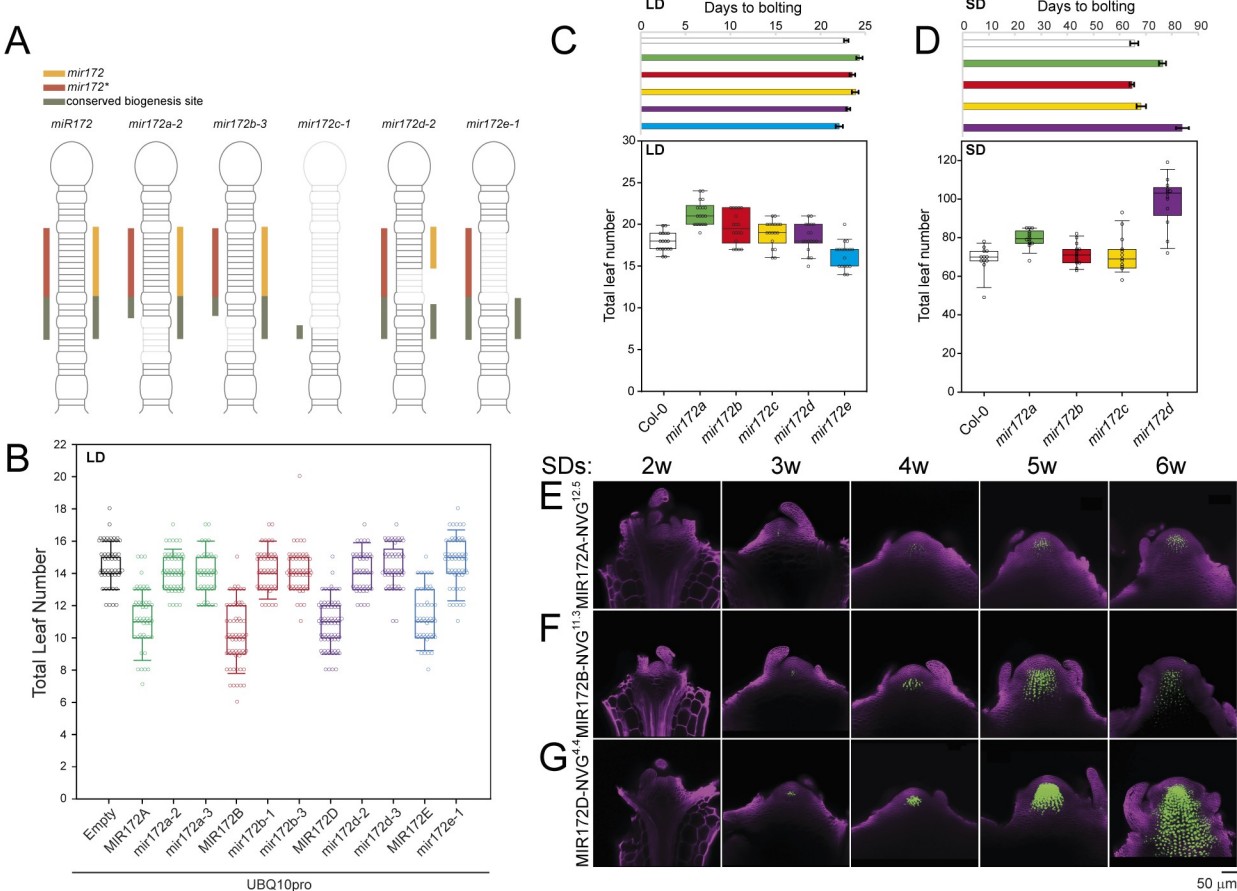

**Fig 2. Functional analysis of single mir172 mutants LDs and SDs.** (A) Schematic depicting the structure of a representative "reference" miR172 precursor and the mutant forms generated using CRISPR-Cas-9 DNA editing. The orange, red, and green bars represent the coding sequences for miR172, miR172*, and the biogenesis site required for DCL1 cleavage, respectively. Truncation of these bars and opaque sections of the mutant mir172 precursor indicates the regions that were deleted via CRISPR-Cas-9 in *mir172a-2*, *mir172b-3*, *mir172c-1*, *mir172d-2*, and *mir172e-1*. (B) A graph depicting the TLN of first generation transformants in a Col-0 background harboring a T-DNA containing only the UBQ10pro-3′OCS promoter-terminator cassette or expressing reference Col-0 or mutated versions of the indicated MIR172 gene under the control of the UBQ10pro-3′OCS cassette. (C, D) Graphs depicting the TLN and bolting times of WT and plants harboring single mir172 mutations in (C) LD and (D) SD conditions. Data underlying panels B–D are provided in S2 Data. (E–G) Confocal laser scanning micrographs of the shoot apices of (E) MIR172A-NVG, (F) MIR172B-NVG, and (G) MIR172D-NVG transgenic plants grown in SD conditions and harvested at the indicated times after germination. Fluorescence from the Venus protein is artificially colored in green, and the fluorescence from the Renaissance dye is artificially colored in magenta. Cas-9, CRISPR associated protein-9; CRISPR, clustered regularly interspaced short palindromic repeats; LD, long-day; NVG, NLS-Venus-GUS; SD, short-day; TLN, total leaf number; WT, wild-type.

Therefore, we generated mutations in each of the 5 *MIR172* genes using previously described CRISPR-Cas-9 systems [52,53]. We inactivated the *MIR172* genes by deleting or severely disrupting the sequences that comprise the miRNA/miRNA* duplex or by disrupting the sequences that are necessary for DICER-LIKE 1 (DCL1)-mediated processing [15,16,54]. The guide RNAs used, their activity, and the sequences of the stably inherited mutations are shown in S4 Fig. We identified 3 classes of mutation: (i) modifications of the stem region beneath the miRNA/miRNA* duplex (*mir172a-2*, *mir172a-3*, *mir172b-1*, and *mir172b-3*); (ii) modifications of the miRNA/miRNA* duplex (*mir172d-2* and *mir172d-3*); and (iii) deletions of the entire miRNA region or the entire miRNA/miRNA* duplex (*mir172c-1* and *mir172e-1*) (Fig 2A, S4M Fig). Notably, the predicted secondary structures of the first class of *mir172* mutant precursors were altered relative to their WT counterparts (S5 Fig), suggesting that their

processing would be impaired [15,16]. To test the effect of the mutations, we ectopically expressed the endogenous and mutant genes *in planta* to assay their effects on flowering time. While the mean total leaf number (TLN) produced by plants ectopically expressing the WT *MIR172* genes was lower than the empty vector control, the mean TLN produced by plants expressing mutated versions was similar to the empty vector control (Fig 2B, Table 1, S1 Table). These data indicated that the *mir172* mutant alleles were strongly impaired in function and may be null alleles, and this was further supported by testing mir172 levels in higher-order mutants (see below).

## Analysis of flowering time behavior of *mir172* mutants

We removed the T-DNA harboring the CRISPR-Cas-9 cassette by genetic crossing and generated quintuple mutants using a selection of these novel *mir172* alleles. We performed whole genome sequencing of the resulting quintuple mutant and found no off-target mutations induced by the CRISPR-Cas-9 cassettes (Materials and methods). We re-isolated all individual single mutants from plants that were segregating for each selected *mir172* mutant and performed flowering time assays in long and short photoperiods. Of the single mutants grown under LD conditions, the average TLN of *mir172a-2* and *mir172b-3* were reproducibly higher relative to the reference WT genotype, Col-0 (Fig 2C, Table 1, S1 Table). In days to bolting (DTB), *mir172a-2* was slightly higher than Col-0 (Fig 2C, Table 1, S1 Table). By contrast, these metrics were mostly comparable between the other *mir172* mutant alleles and Col-0 (Fig 2C, Table 1, S1 Table). Therefore, *MIR172A* and *MIR172B* are the most important in floral promotion under LDs.

Under SD conditions, *mir172a-2* also flowered later than Col-0, whereas *mir172b-3* and *mir172c-1* flowered at a time similar to Col-0 (Fig 2D, Table 1, S1 Table). In contrast to LD conditions, *mir172d-3* flowered much later than Col-0 when grown under SD conditions (Fig 2D, Table 1, S1 Table). These results suggested that miR172 activity is more important in the absence of inductive photoperiods and that *MIR172D* contributes to a large degree of this activity. Therefore, we tested the expression of our *MIR172* reporters under these conditions (Fig 2E–2G, S6 Fig) and found that the expression patterns in the shoot apex were similar to those observed under LDs. However, expression of *MIR172A-NVG* was not detectable until 2 to 3 weeks of growth under SDs, compared with approximately 10d under LDs (Fig 2E, S6A Fig). Expression of *MIR172B-NVG* and *MIR172D-NVG* was also not detectable until 3 weeks of growth in SDs (Fig 2F and 2G, S6B, S6C–S6F Fig), compared to 10 to 12 LDs.

We then analyzed the flowering time behavior of *mir172* mutant combinations and observed progressively later flowering under both LD and SD conditions as more *MIR172* genes were inactivated (Fig 3A and 3B). The *mir172abcd* quadruple mutant flowered later than the *mir172abc* triple mutant in both conditions, but it was substantially later under SDs, consistent with an important role for *MIR172D* under these conditions (Fig 3B, Table 1, S1 Table). The leaf number and DTB were comparable between quintuple *mir172abcde* mutant plants and quadruple *mir172abcd* mutant plants that retained a functional *MIR172E* allele, consistent with a weak role for this gene in controlling flowering time (Fig 3B). Furthermore, both quadruple and quintuple mutants grown under short photoperiods initially flowered from axillary shoots rather than the primary shoot (70/77 plants), indicating that miR172 is essential for the earlier timing of the floral transition at the SAM compared to axillary meristems under these conditions (Fig 3C). Siliques of quadruple and quintuple mutant plants exhibited a swollen shape, indicative of defective floral meristem termination (S7A Fig). The severity of this phenotype decreased acropetally, which mirrored previous reports of the severity of *ap2* single mutant phenotypes increasing acropetally [55]. Plants lacking *MIR172A*, *MIR172B*, and

**Table 1. Flowering time experiments and statistical groupings.**

| | Rosette leaves | Cauline leaves | Total leaves | Days | *N* |
|---|---|---|---|---|---|
| **Experiment 1** | | | | | |
| Long days | | | | | |
| UBQ10pro:CTRL | 11.35 ± 1.07* | 3.28 ± 0.57* | 14.63 ± 1.21[ab] | N.A. | 65 |
| UBQ10pro:MIR172A | 8.67 ± 1.48* | 2.65 ± 0.67* | 11.33 ± 1.76[c] | N.A. | 55 |
| UBQ10pro:mir172a-2 | 10.98 ± 0.91* | 2.92 ± 0.61* | 13.90 ± 1.31[a] | N.A. | 48 |
| UBQ10pro:mir172a-3 | 11.06 ± 0.91* | 3.03 ± 0.5k3* | 14.09 ± 1.12[ab] | N.A. | 64 |
| UBQ10pro:MIR172B | 7.51 ± 1.52* | 2.60 ± 0.60* | 10.10 ± 1.87[c] | N.A. | 67 |
| UBQ10pro:mir172b-1 | 11.04 ± 1.09* | 3.13 ± 0.59* | 14.17 ± 1.27[ab] | N.A. | 53 |
| UBQ10pro:mir172b-3 | 11.11 ± 1.09* | 3.05 ± 0.45* | 14.16 ± 1.29[ab] | N.A. | 64 |
| UBQ10pro:MIR172D | 8.11 ± 1.28* | 2.72 ± 0.55* | 10.83 ± 1.46[c] | N.A. | 82 |
| UBQ10pro:mir172d-2 | 10.96 ± 0.97* | 3.08 ± 0.53* | 14.04 ± 1.21[ab] | N.A. | 50 |
| UBQ10pro:mir172d-3 | 11.33 ± 0.99* | 3.11 ± 0.52* | 14.44 ± 1.27[ab] | N.A. | 57 |
| UBQ10pro:MIR172E | 8.35 ± 1.43* | 2.96 ± 0.66* | 11.31 ± 1.61[c] | N.A. | 51 |
| UBQ10pro:mir172e-1 | 11.56 ± 1.11* | 3.10 ± 0.62* | 14.66 ± 1.45[b] | N.A. | 62 |
| **Experiment 2.1** | | | | | |
| Long days | | | | | |
| Col-0 | 14.72 ± 0.83a | 3.17 ± 0.79ad | 17.89 ± 1.23[a] | 22.89 ± 1.02[ab] | 18 |
| *mir172a-2* | 16.56 ± 1.50b | 4.83 ± 0.79b | 21.39 ± 1.46[b] | 24.33 ± 1.37[a] | 18 |
| *mir172b-3* | 15.89 ± 1.60ab | 3.72 ± 0.67[bd] | 19.61 ± 1.91[bc] | 23.56 ± 1.46[ab] | 18 |
| *mir172c-1* | 15.56 ± 1.69ac | 3.39 ± 0.50[bd] | 18.94 ± 1.55[ac] | 23.89 ± 1.60a | 18 |
| *mir172d-3* | 14.56 ± 1.50ac | 3.89 ± 0.58d | 18.44 ± 1.72[ac] | 23.11 ± 1.02[ab] | 18 |
| *mir172e-1* | 13.44 ± 1.34[c] | 2.89 ± 0.58a | 16.33 ± 1.57[d] | 22.11 ± 1.02b | 18 |
| **Experiment 2.2** | | | | | |
| Long days | | | | | |
| Col-0 | 10.14 ± 1.23a | 3.29 ± 0.61a | 13.43 ± 1.70[a] | 21.79 ± 0.80[a] | 14 |
| *mir172c-1* | 10.13 ± 0.99a | 3.33 ± 0.72a | 13.47 ± 1.30[a] | 21.80 ± 0.77[a] | 15 |
| *mir172e-1* | 9.67 ± 1.29a | 3.07 ± 0.70a | 12.73 ± 1.75[a] | 21.80 ± 0.68[a] | 15 |
| *mir172c-1 e-1* | 10.33 ± 1.11a | 3.00 ± 0.53a | 13.33 ± 1.45[a] | 21.40 ± 0.63[a] | 15 |
| **Experiment 2.3** | | | | | |
| Long days | | | | | |
| Col-0 | 12.91 ± 1.23a | 3.27 ± 0.46a | 16.18 ± 1.44[a] | 23.36 ± 1.53[a] | 22 |
| *mir172a-2* | 15.64 ± 1.36[ab] | 4.82 ± 0.96b | 20.45 ± 1.99[ab] | 24.64 ± 1.89[ab] | 22 |
| *mir172a-2 b-3* | 18.18 ± 2.20bc | 5.27 ± 0.88bc | 23.45 ± 2.63bc | 25.73 ± 1.52[b] | 22 |
| *mir172a- b-3 c-1* | 20.20 ± 2.75[c] | 5.35 ± 0.75bc | 25.55 ± 3.30[c] | 26.45 ± 2.19[bc] | 20 |
| *mir172a- b-3 c-1 d-3* | 25.09 ± 2.83d | 5.64 ± 1.36bc | 30.73 ± 3.09[d] | 28.27 ± 2.05[cd] | 22 |
| *mir172a- b-3 c-1 d-3 e-1* | 25.61 ± 1.80d | 6.17 ± 1.47c | 31.78 ± 2.71[d] | 29.61 ± 1.75[d] | 23 |
| **Experiment 2.4** | | | | | |
| Long days | | | | | |
| Col-0 | 15.33 ± 1.88a | 3.06 ± 0.80a | 18.39 ± 2.25[a] | 23.67 ± 1.24[a] | 18 |
| *mir172a-2 b-3 d-2* | 27.11 ± 1.71b | 7.44 ± 0.62b | 34.56 ± 1.98[b] | 31.11 ± 2.08[b] | 18 |
| *mir172a-2 b-3 c-1 d-2 e-1* | 28.83 ± 1.72b | 6.22 ± 0.73c | 35.06 ± 1.92[b] | 32.83 ± 1.92[b] | 18 |
| 35Spro:MIM172 | 23.72 ± 2.05c | 5.39 ± 0.98d | 29.11 ± 2.0[c] | 38.11 ± 1.45[c] | 18 |
| **Experiment 2.5** | | | | | |
| Long days | | | | | |
| Col-0 | 15.17 ± 0.62a | 2.78 ± 0.43a | 17.94 ± 0.80[a] | 24.83 ± 0.62[a] | 18 |
| *ap2-12* | 13.61 ± 0.61b | 2.28 ± 0.46a | 15.89 ± 0.83[b] | 21.56 ± 1.25[b] | 18 |
| *mir172a-2 b-3 d-2* | 26.89 ± 1.02c | 6.06 ± 0.87b | 32.94 ± 1.39[c] | 35.17 ± 0.38[c] | 18 |

(*Continued*)

**Table 1.** (Continued)

| | Rosette leaves | Cauline leaves | Total leaves | Days | N |
|---|---|---|---|---|---|
| mir172a- b-3 d-2 ap2-12 | 22.56 ± 0.92d | 1.17 ± 0.38c | 23.72 ± 0.89[d] | 32.61 ± 0.50[d] | 18 |
| **Experiment 3.1** | | | | | |
| Short days | | | | | |
| Col-0 | 60.67 ± 6.69a | 8.33 ± 0.98a | 69.00 ± 7.16[a] | 65.58 ± 5.18a | 12 |
| mir172a-2 | 71.79 ± 4.44b | 7.86 ± 1.96a | 79.64 ± 4.57[b,] | 76.36 ± 4.91[b] | 14 |
| mir172b-3 | 63.06 ± 4.78a | (8.67 ± 1.11)a | 71.19 ± 5.34[a] | 64.75 ± 3.57[a] | 16 (15) |
| mir172c-1 | 62.50 ± 8.79a | 8.63 ± 0.89a | 71.13 ± 9.02[a] | 68.19 ± 7.37[a] | 16 |
| mir172d-3 | 79.00 ± 9.69b | 20.69 ± 5.09b | 98.21 ± 12.68[b] | 83.86 ± 9.19[b] | 14 (13) |
| **Experiment 3.2** | | | | | |
| Short days | | | | | |
| Col-0 | 52.22 ± 4.17a | 8.00 ± 1.24* | 60.22 ± 4.65* | 58.94 ± 4.98[a] | 18 |
| mir172a-2 | 61.53 ± 4.47[ab] | 9.27 ± 1.10* | 70.80 ± 4.16* | 64.87 ± 4.21[ab] | 15 |
| mir172a-2 b-3 | 76.95 ± 8.59bc | 8.84 ± 0.76* | 85.79 ± 8.59* | 71.47 ± 5.32[bc] | 19 |
| mir172a- b-3 c-1 | 81.70 ± 8.71c | 9.55 ± 0.89* | 91.25 ± 8.74* | 76.35 ± 7.07[c] | 20 |
| mir172a- b-3 c-1 d-3 | 129.14 ± 7.02[d,¥] | N.A. | N.A. | (150.28 ± 21.37)[d] | 21 (18) |
| mir172a- b-3 c-1 d-3 e-1 | 131.00 ± 9.21[d,¥] | N.A. | N.A. | (142.64 ± 15.02)[d] | 26 (25) |
| **Experiment 3.3** | | | | | |
| Short days | | | | | |
| Col-0 | 57.1 ± 3.74a | 7.95 ± 1.15* | 65.05 ± 3.65* | 59.20 ± 2.65[a] | 20 |
| mir172a-2 b-3 d-2 | 162.55 ± 6.44b | N.A. | N.A. | 141.7 ± 8.64[b] | 20 |
| mir172a- b-3 c-1 d-3 e-1 | 165.1 ± 8.06b | N.A. | N.A. | 142.4 ± 13.54[b] | 20 |
| 35Spro:MIM172 | 122.75 ± 5.89c | N.A. | N.A. | (128.61 ± 7.10)[c] | 20 (18) |
| **Experiment 3.4** | | | | | |
| Short days | | | | | |
| Col-0 | 57.1 ± 3.74a | 7.95 ± 1.15* | 65.05 ± 3.65* | 59.20 ± 2.65[a] | 20 |
| ap2-12 | 39.85 ± 3.65b | 4.15 ± 0.93* | 44.00 ± 3.8* | 53.60 ± 1.93[b] | 20 |
| mir172a-2 b-3 d-2 | 159.45 ± 6.05c | N.A. | N.A. | 140.2 ± 9.99[c] | 20 |
| ap2-12 mir172a- b-3 d-2 | 135.9 ± 6.70d | N.A. | N.A. | 118.7 ± 6.59[d] | 20 |
| **Experiment 4.1** | | | | | |
| Long days | | | | | |
| Col-0 | 10.00 ± 1.28 a | 3.08 ± 0.29a | 13.08 ± 1.44[a] | 29.33 ± 0.78[a] | 12 |
| spl15-1 | 14.75 ± 1.48b | 3.08 ± 0.67a | 17.83 ± 1.90[b] | 32.25 ± 1.76[b] | 12 |
| mir172a-2 b-3 | 15.50 ± 2.24b | 4.83 ± 1.03b | 20.33 ± 3.14[b] | 34.00 ± 2.22[bc] | 12 |
| spl15-1 mir172a-2 b-3 | 22.75 ± 5.38c | 5.50 ±1.17b | 28.25 ± 6.33[c] | 36.83 ± 3.24[c] | 12 |
| **Experiment 4.2** | | | | | |
| Long days | | | | | |
| Col-0 | 10.77 ± 1.39a | 3.15 ± 0.54a | 13.92 ± 1.67[a] | 30.04 ± 1.56[a] | 26 |
| spl15-1 | 12.54 ± 1.77b | 3.42 ±0.58a | 15.96 ± 2.11[b] | 31.31 ± 2.40[ab] | 26 |
| ful-2 | 11.40 ± 1.32[ab] | 4.60 ±0.71b | 16.00 ± 1.61[b] | 32.72 ± 1.67[b] | 25 |
| spl15-1 ful-2 | 17.40 ± 3.07c | 8.35 ± 1.83c | 25.77 ± 4.73[c] | 37.04 ± 2.11[c] | 26 |
| **Experiment 4.3** | | | | | |
| Long days | | | | | |
| Col-0 | 12.29 ± 1.31a | 3.62 ± 0.67a | 15.90 ± 1.41[a] | (30.05 ± 1.54)[a] | 21 (20) |
| ful-2 | 11.62 ± 1.48a | 5.00 ± 0.63b | 16.62 ± 1.83[a] | 32.57 ± 1.80[ab] | 21 |
| ful-2 mir172a-2 | 15.71 ± 4.06b | (6.41 ± 1.28)c | 21.78 ± 5.24[b] | 37.28 ± 4.17[cd] | 18 (17) |
| ful-2 mir172b-3 | 16.19 ± 3.60b | 4.81 ± 0.87b | 21.00 ± 3.92[b] | 34.19 ± 2.71[c] | 21 |
| ful-2 mir172a-2 b-3 | 21.20 ± 3.12c | 7.00 ± 1.17c | 28.20 ± 3.91[c] | 40.40 ± 4.16[bd] | 20 |

(*Continued*)

**Table 1.** (*Continued*)

| | Rosette leaves | Cauline leaves | Total leaves | Days | *N* |
|---|---|---|---|---|---|
| **Experiment 4.4** | | | | | |
| Long days | | | | | |
| Col-0 | 10.67 ± 1.28a | 2.89 ± 0.58[a] | 13.56 ± 1.54[a] | 31.00 ± 1.73[a] | 18 |
| *ful-2* | 9.85 ± 1.07a | 4.00 ± 0.71[b] | 13.85 ± 1.57[a] | 34.38 ± 2.29[b] | 13 |
| *mir172a-2 b-3 c-1 d-3* | 21.71 ± 2.37b | 5.48 ± 0.75[c] | 27.19 ± 2.46[b] | 40.33 ± 2.11[c] | 21 |
| *ful-2 mir172a-2 b-3 c-1 d-3* | 29.11 ± 3.56c | 16.44 ± 5.12[d] | 45.56 ± 7.73[c] | 47.17 ± 3.68[d] | 18 |
| **Experiment 5.1** | | | | | |
| Long days | | | | | |
| Col-0 | 9.39 ± 0.61a | 2.72 ± 0.57[a] | 12.11 ± 1.02[a] | 23.00 ± 1.50[a] | 18 |
| rSPL15 | 5.65 ± 1.11b | 3.76 ± 0.75[b] | 9.41 ± 1.12[b] | 21.24 ± 0.90[b] | 17 |
| *ful-2 mir172a-2 b-3* | 18.88 ± 2.06c | 7.00 ± 0.97[e] | 25.88 ± 2.60[c] | 34.31 ± 2.57[c] | 16 |
| rSPL15 *ful-2* | 6.33 ± 0.72be | 5.60 ± 1.06[c] | 11.93 ± 1.39[a] | 23.13 ± 1.06[a] | 15 |
| rSPL15 *mir172a-2 b-3* | 7.25 ± 0.86de | 4.19 ± 1.05[b] | 11.44 ± 1.09[a] | 25.25 ± 1.06[d] | 16 |
| rSPL15 *ful-2 mir172a-2 b-3* | 8.28 ± 1.36d | 8.67 ± 1.37[d] | 16.94 ± 2.39[d] | 26.28 ± 1.53[d] | 18 |
| **Experiment 5.2** | | | | | |
| Short days | | | | | |
| Col-0 | 55.08 ± 2.07a | N.A. | N.A. | 52.83 ± 2.04[a] | 12 |
| rSPL15 | 16.67 ± 2.31b | N.A. | N.A. | 35.58 ± 2.11[b] | 12 |
| *ful-2 mir172a-2 b-3* | 84.67 ± 7.02d | N.A. | N.A. | 69.58 ± 4.01[d] | 12 |
| rSPL15 *ful-2* | 26.33 ± 5.31c | N.A. | N.A. | 48.42 ± 5.92[ac] | 12 |
| rSPL15 *mir172a-2 b-3* | 25.83 ± 7.80bc | N.A. | N.A. | 45.58 ± 5.38[c] | 12 |
| rSPL15 *ful-2 mir172a-2 b-3* | 37.42 ± 10.05c | N.A. | N.A. | 59.17 ± 10.36[ad] | 12 |
| **Experiment 6.1** | | | | | |
| Short days | | | | | |
| Col-0 | 51.69 ± 5.72a | 8.23 ± 0.73a | 59.92 ± 5.66a | (56.50 ± 2.11)a | 13 (12) |
| *ful-2* | 47.36 ± 4.96a | 17.93 ± 5.00bc | 65.29 ± 8.50a | 72.36 ± 5.85b | 14 |
| *mir172a-2 b-3* | 74.92 ± 7.15bc | 9.23 ± 1.17a | 84.15 ± 7.61b | 76.31 ± 6.90bc | 13 |
| *ful-2 mir172a-2* | 63.86 ± 8.61d | 24.86 ± 4.29b | 88.71 ± 10.94bc | 80.79 ± 8.21cd | 14 |
| *ful-2 mir172b-3* | 72.64 ± 15.68[bd] | (22.50 ± 4.44)b | (93.42 ± 13.15)bcd | 96.21 ± 13.69e | 14 (12) |
| *ful-2 mir172a-2 b-3* | 77.00 ± 10.95bc | 24.79 ± 2.39b | 101.79 ± 10.56d | 91.43 ± 7.40e | 14 |
| *spl15-1* | 90.14 ± 8.50[c] | (10.58 ± 1.78)ac | (98.83 ± 7.28)cd | 88.14 ± 9.73de | 14 (12) |
| **Experiment 6.2** | | | | | |
| Short days | | | | | |
| Col-0 | N.A. | N.A. | N.A. | 98.15 ± 12.17a | 48 |
| *ful-2* | N.A. | N.A. | N.A. | 119.29 ± 19.35bc | 45 |
| *spl15-1* | N.A. | N.A. | N.A. | 135.09 ± 13.93de | 47 |
| *mir172a-2* | N.A. | N.A. | N.A. | 106.46 ± 15.44a | 48 |
| *mir172b-3* | N.A. | N.A. | N.A. | 100.81 ± 15.9a | 47 |
| *mir172a-2 b-3* | N.A. | N.A. | N.A. | 107.56 ± 19.23ac | 50 |
| *ful-2 mir172a-2* | N.A. | N.A. | N.A. | 122.92 ± 14.62[bd] | 39 |
| *ful-2 mir172b-3* | N.A. | N.A. | N.A. | 155.83 ± 20.55[efg] | 18 |
| *ful-2 mir172a-2 b-3* | N.A. | N.A. | N.A. | 136.53 ± 17.79[def] | 34 |
| *spl15-1 mir172a-2* | N.A. | N.A. | N.A. | 157.82 ± 14.54[fg] | 28 |
| *spl15-1 mir172b-3* | N.A. | N.A. | N.A. | 170.57 ± 13.96[g] | 28 |

(*Continued*)

**Table 1.** (Continued)

| | Rosette leaves | Cauline leaves | Total leaves | Days | N |
|---|---|---|---|---|---|
| *spl15-1 ful-2* | N.A. | N.A. | N.A. | $190.00 \pm 21.57^{\text{g}}$ | 11 |

Groups are assigned by superscript letters, a–g, at α < 0.01. Figures in brackets correspond to the observed number of plants for that metric, which is also indicated in brackets.

*Not assessed.

¥Leaf counting stopped approximately 30 days after other plants had flowered.

LD, long-day; SD, short-day; WT, wild-type.

*MIR172D* activity also bore severely serrated cauline leaves (S7B Fig), which were less serrated in *mir172abc* or *mir172ab* mutants and very similar to Col-0 in the single mutant combinations. Supported by the imaging and RT-qPCR results presented above, these results suggest that *MIR172E* plays a very minor role during the floral transition and floral organ development.

With the exception of the comparison between quadruple and quintuple mutants, the smallest effect size in terms of leaf number and DTB was observed when comparing *mir172abc* and *mir172ab* mutants (Fig 3A and 3B, Table 1, S1 Table). The TLN and DTB of *mir172c-1 e-1* double mutants were similar to Col-0 under LDs (S7C Fig, Table 1, S1 Table). Furthermore, all *mir172abc* plants grown under SD conditions flowered from the main shoot and bore siliques of WT appearance. This suggests that although *MIR172C* may play some role in controlling flowering time, its contribution is small when compared to *MIR172A*, *MIR172B*, and *MIR172D*.

The TLN and DTB of *mir172ab* were clearly higher than Col-0 in both photoperiods (Fig 3A and 3B, Table 1, S1 Table). Although the differences between *mir172a* single mutants and Col-0 were similar to our previous results (Fig 2C and 2D, Table 1, S1 Table), the variation was increased in these experiments (Fig 3A and 3B, Table 1, S1 Table). We noted that the number of leaves produced by *mir172a* mutants often appeared greater relative to Col-0 than the differences in DTB. Therefore, we tested the leaf initiation rate of *mir172* mutants and found that the plastochron was longer than Col-0, with robust differences arising after 40 d of growth under short photoperiods (S7D Fig). Therefore, counting the leaf number of *mir172* mutants alone is not sufficient to assess their flowering times.

Taken together, these genetic data supported the idea that *MIR172A*, *MIR172B*, and *MIR172D* are the main regulators of flowering time in the *MIR172* gene family. This is in agreement with the expression patterns of the reporter constructs, with only *MIR172A-NVG*, *MIR172B-NVG*, and *MIR172D-NVG* being detectably expressed in the shoot apex prior to and around the time of the floral transition. To test this further, we produced *mir172a-3 b-3 d-2* triple mutants and compared their flowering time behaviors under long and SDs with the *mir172abcde* quintuple mutants generated above and the previously described *35Spro: MIM172* transgenic line, which ectopically expresses a target mimicry that depletes miR172 activity [5] (Fig 3D–3F). As observed previously, *mir172abcde* plants flowered much later than Col-0 (Fig 3D–3F). The leaf numbers and DTB of *mir172abd* were very similar to *mir172abcde* under both conditions (Fig 3D and 3E, Table 1, S1 Table), with most *mir172abd* and *mir172-abcde* plants initially flowering from axillary shoots when grown under short photoperiods. The leaf numbers and DTB of *35Spro:MIM172* transgenic lines were lower than both the triple and quintuple mutants under LDs and SDs (Fig 3D and 3E, Table 1, S1 Table). Furthermore, the morphology of the *35Spro:MIM172* rosette leaves was markedly different to any of the *mir172* mutant plants, and the rosette diameter was substantially smaller, which was particularly noticeable in SDs (S7F Fig). We conclude that *MIR172A*, *MIR172B*, and *MIR172D* are the

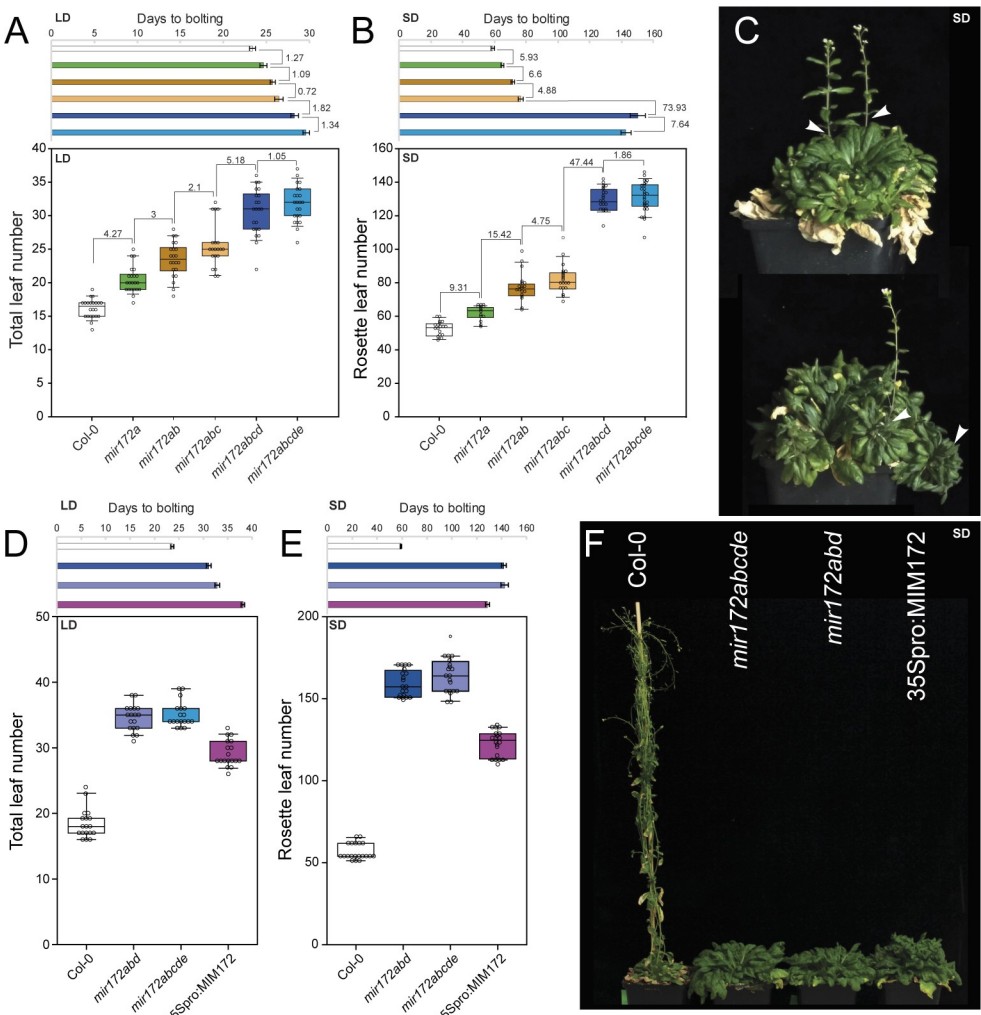

**Fig 3. Flowering time of higher-order *mir172* mutants in LDs and SDs.** (A, B) Graphs depicting DTB and TLN or RLN of Col-0, *mir172a-2*, *mir172a-2 b-3*, *mir172a-2 b-3 c-1*, *mir172a-2 b-3 c-1 d-3*, and *mir172a-2 b-3 c-1 d-3 e-1* mutant combinations in LD (A) and SD (B) conditions. The difference between the means of the indicated genotypes is shown above the box plots. Data underlying panels (A) and (B) are provided in S2 Data. (C) Photographs of *mir172a-2 b-3 c-1 d-3 e-1* plants after approximately 6 months of growth under SD conditions. Plant initiated flowering shoots from axillary meristems (white arrowheads) rather than the primary meristem. (D, E) Graphs depicting DTB and total or RLN of Col-0, *mir172a-2 b-3 d-2*, *mir172a-2 b-3 c-1 d-3 e-1* mutant combinations and *35Spro:MIM172* in LD (D) and SD (E) conditions. Data underlying panels (D) and (E) are provided in S2 Data. (F) A photograph of Col-0, *mir172a-2 b-3 d-2*, *mir172a-2 b-3 c-1 d-3 e-1* mutant combinations and *35Spro:MIM172* grown under SD conditions. DTB, days to bolting; LD, long-day; RLN, Rosette leaf number; SD, short-day; TLN, total leaf number.

primary regulators of flowering time in the *MIR172* family. We also conclude that *miR172* action is essential for flowering from the SAM under noninductive photoperiods and strongly delays flowering, while under inductive photoperiods, it plays a less important role in flowering, and this can be eventually bypassed.

## Activity of *miR172* in the leaves and meristems under inductive photoperiods

Under inductive LD photoperiods, gene activity in leaves and shoot apices make major contributions to floral induction; therefore, to understand the roles of the *MIR172* genes under these

conditions, we examined the expression of the *MIR172-NVG* reporters in the leaves of plants grown under LDs. For this analysis, we used histological staining with the β-glucuronidase enzyme (GUS), which comprises part of our NVG reporters (S1, S8 and S9 Figs). Strong GUS enzyme activity was detected in the cotyledons and throughout the leaves of *MIR172B-NVG* plants, which grew weaker as the leaf aged. In contrast, *MIR172A-NVG* activity was weakly but reproducibly detected in the vasculature of young late-arising rosette and/or cauline leaves (approximately 21 d), and *MIR172C-NVG* was sometimes detected in a similar pattern (Fig 4, S8 and S9 Figs). *MIR172D-NVG* expression was not reliably detected in leaves (Fig 4, S8 and S9 Figs). These patterns are largely in agreement with a small RNA sequencing (RNA-seq) dataset in which the abundance of the *miR172a-b* isoform was 3 orders of magnitude higher in the leaves than the *miR172c-d* or *miR172e* isoforms (S9O Fig) [56].

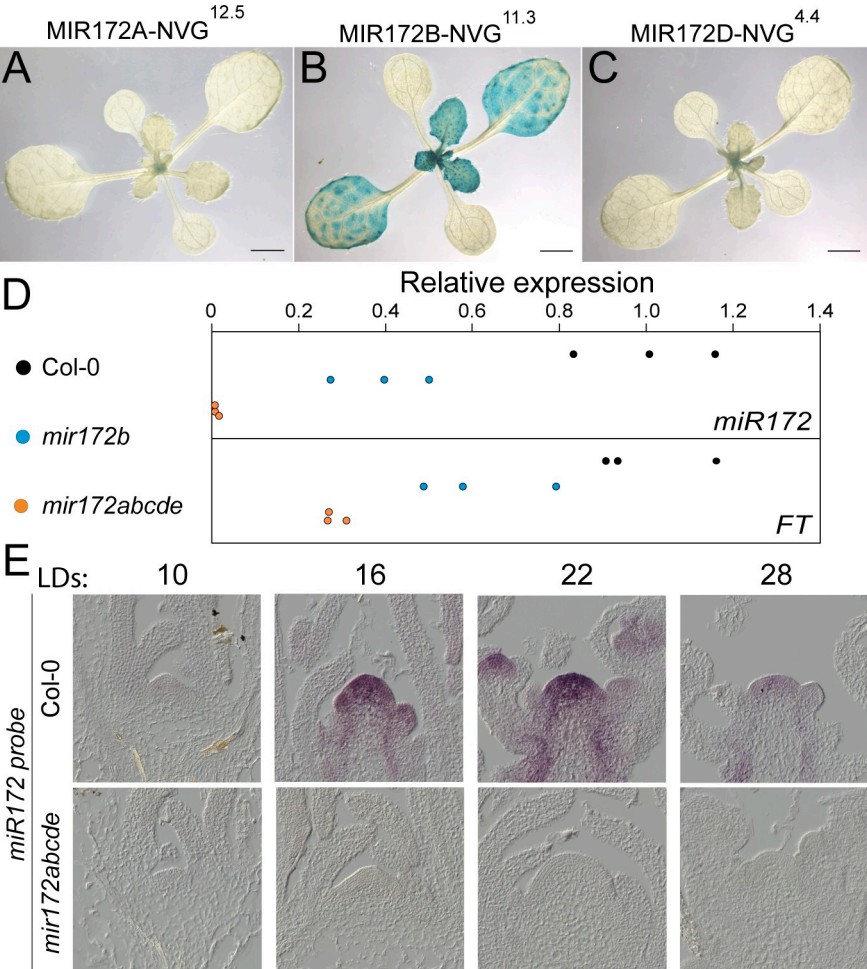

**Fig 4. Activity of *MIR172* genes in leaves.** (A–C) Photographs of 14-day-old (A) *MIR172A-NVG*, (B) *MIR172B-NVG*, and (C) *MIR172D-NVG* transgenic plants grown in LD conditions after GUS staining and clearing. Scale bar indicates 2 mm. (D) Graphs depicting RT-qPCR results of miR172 (upper panel) and *FT* mRNA (lower panel) in leaves 1 and 2 of Col-0, *mir172b-3* and *mir172a-2 b-3 c-1 d-3 e-1* grown for 12 LDs, harvested at ZT 16. Each point represents an individual biological replicate. Each biological replicate was normalized to the mean of the 3 Col-0 biological replicates. Data underlying panels D are provided in S1 Data. (E) Micrographs of RNA *in situ* hybridizations using a commercial probe designed to recognize miR172 of Col-0 (upper panels) and *mir172a-2 b-3 c-1 d-3 e-1* (lower panels) apices grown under LD conditions for the indicated times. FT, FLOWERING LOCUS T; GUS, β-glucuronidase enzyme; LD, long-day; NVG, NLS-Venus-GUS; RT-qPCR, quantitative reverse transcription PCR; ZT, zeitgeber time.

Together, these data indicate that *MIR172B* makes a major contribution to the pool of miR172 present in the leaves. To quantify the amount of miR172 present, we performed RT-qPCR on mature miR172 in Col-0, *mir172b* single mutants, and *mir172abcde* quintuple mutants. *MIR172B* contributed 50% to 73% of total miR172 levels in leaves 1 and 2 of 12-day-old LD-grown plants (2-tailed paired *t* test *p* = 0.021). We did not test miR172 levels in the other single mutants, but in leaves of *mir172abcde*, quintuple mutants miR172 levels were at approximately 1% of those in Col-0 (2-tailed paired *t* test *p* = 0.009) (Fig 4D). This quantitative result is consistent with the absence of miR172 signal in the shoot apices of quintuple mutant plants, as determined by RNA *in situ* hybridization both under LDs and SDs (Fig 4E, S7E Fig). Some of the AP2-LIKE TFs whose mRNAs are miR172 targets are thought to directly suppress the transcription of the florigen-encoding *FT* gene in leaves [7,22,27]. We therefore tested *FT* mRNA levels in the same leaf material of these genetic backgrounds and found that *FT* mRNA levels were between 49% and 79% of Col-0 levels in *mir172b* mutants (2-tailed paired *t* test *p* = 0.008) (Fig 4D). In *mir172abcde* quintuple mutants, *FT* mRNA levels were between 27% and 31% of Col-0 levels (2-tailed paired *t* test *p* = 0.015), which suggests that reduced FT partially contributes to the late-flowering phenotype of LD-grown *mir172* quintuple mutants (Fig 4D). However, sufficient *FT* and *TSF* expression may be retained to facilitate some response to LDs, as suggested by the dramatic difference between *miR172abcde* mutant flowering in LD- and SD-grown plants (Fig 2A and 2B).

## Interplay of *AP2* and *MIR172* during the floral transition

MiR172 inhibits the activity of AP2-like proteins either by inducing degradation of their mRNAs or inhibiting their translation [8,22,45]. Unfortunately, the activities and the mRNA or protein expression profiles of most of these *AP2-LIKE* genes have not been precisely determined. Furthermore, two of the best characterized *AP2-LIKE* genes, *TOE1* and *SMZ*, are genetically linked to *MIR172A* and *MIR172D*, respectively, complicating genetic analysis (S10 Fig). Moreover, little is known about *TOE2* and *SNZ* activity, and the main function of *TOE3* is probably during floral organ formation [24]. Therefore, we initially focused on testing the relationship between *AP2* and miR172 during floral transition. Previous work that included genome-wide assessments of AP2 activity suggested it exerts the majority of its influence on flowering time at the shoot apex [21], although it appears to be expressed throughout the plant [57]. Furthermore, RNA *in situ* hybridization showed that *AP2* mRNA is highly abundant in the vegetative SAM but reduced in the inflorescence meristem [31], which is consistent with its role as a negative regulator of flowering time. To understand how the expression pattern of AP2 progresses during the floral transition, we generated a fluorescently tagged version of AP2 driven by its endogenous regulatory sequences (S11 Fig). We then assessed the expression pattern of AP2-Venus in Col-0 by confocal laser microscopy under LDs (Fig 5A). AP2-Venus was present in the shoot meristem at 14 days after germination but at later time points, as the floral transition proceeded, AP2-VENUS was depleted from the apex. This pattern therefore was anticorrelated with the expression of the *MIR172-NVG* reporters (Fig 1). We then introgressed the AP2-VENUS transgene into the *mir172abd* triple mutant background and monitored the expression of AP2-VENUS in LDs (Fig 5A). AP2-VENUS was not detected in the *mir172abd* plants at 14 days after germination, but the signal appeared at 17 LDs and became strongly expressed in a broader domain at 19 LDs (Fig 5A). At 21 LDs, AP2-VENUS was no longer detected at the shoot meristem. This experiment indicated that in *mir172abd* plants the expression of AP2-VENUS protein at the shoot apex is delayed and that it then persists later in development than in Col plants. Interestingly, similar observations were made through *in situ* hybridizations with *AP2* mRNA, as it was absent from the apices of WT plants by 16 LDs but

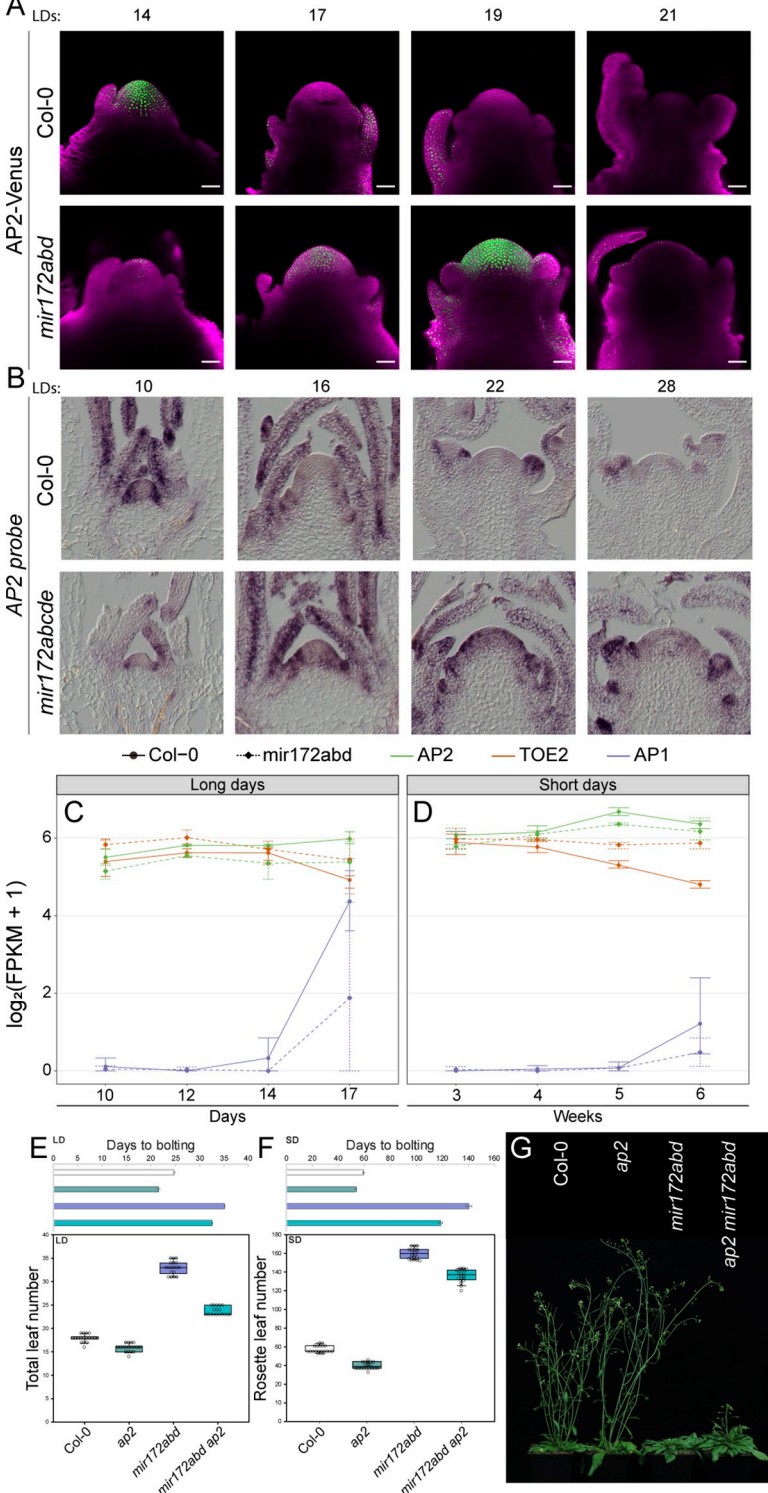

**Fig 5. AP2 activity in the shoot apex.** (A) Confocal laser scanning micrographs of the shoot apices of AP2-Venus transgenic plants grown in LD conditions in a reference Col-0 background (upper panels) and *mir172a-2 b-3 d-2* background (lower panels). Fluorescence from the Venus protein is artificially colored in green, and the fluorescence from the Renaissance 2200 dye is artificially colored in magenta. (B) Micrographs of RNA *in situ* hybridizations using a commercial probe designed to recognize *AP2* mRNA of Col-0 (upper panels) and *mir172a-2 b-3 c-1 d-3 e-1* (lower panels) apices grown under LD conditions for the indicated times. (C, D) Graphs depicting RNA-seq-derived data of

the mRNA levels in apices of *AP2*, *TOE2*, and *AP1* in Col-0 and *miR172abd* at the indicated time points after germination in (C) LDs and (D) SDs. Error bars represent the standard deviation of 3 biological replicates. Data underlying panels (C) and (D) are provided in S1 Data. (E, F) Graphs depicting the DTB and T/RLNs of Col-0, *ap2-12*, *miR172a-2 b-3 d-2*, and *mir172a-2 b-3 d-2 ap2-12* plants grown under (E) LD and (F) SD conditions. Data underlying panels (E) and (F) are provided in S2 Data. (G) Photograph of Col-0, *ap2-12*, *miR172a-2 b-3 d-2*, and *ap2-12 miR172a-2 b-3 d-2* plants grown in LD conditions. AP2, APETALA2; DTB, days to bolting; LD, long-day; RLN, Rosette leaf number; SD, short-day; TLN, total leaf number.

persisted in *mir172abcde* mutants until at least 22 days (Fig 5B). To provide a broader view of the effect of *mir172abd* mutations on the abundance of the mRNAs of *AP2* and the *AP2-L* genes, we performed an RNA-seq analysis of apices of Col-0 and *mir172abd* plants grown under LDs and SDs (Fig 5C and 5D, S11 Fig, S3 Table). At this level of resolution, the abundance of the mRNAs of *AP2* and most *AP2-L* genes in *mir172abd* plants was similar to Col-0 throughout the time course. However, the mRNA of *TOE2* was significantly reduced in Col-0 compared to *mir172abd* at later time points under LDs and SDs. This result is consistent with a previous analysis that found that in plants overexpressing miR172, *TOE2* mRNA was strongly reduced, while the mRNA levels of *AP2* and other *AP2-L* genes were not significantly altered, perhaps due to impaired feedback regulation [45].

The extended duration of expression of AP2-VENUS in *mir172abd* plants indicated that the persistence of AP2 in the SAMs of *mir172abd* plants might contribute to their later flowering. To test this directly, we introduced the null *ap2-12* allele into *mir172abd* triple mutants and scored their flowering time under long and short photoperiods (Fig 5E and 5F). As previously reported, *ap2-12* produced fewer leaves and bolted earlier than Col-0 under both LDs and SDs (Fig 5E–5G, Table 1, S1 Table) [21]. Consistent with this, the number of leaves and bolting time of *ap2 miR172abd* quadruple mutants were much reduced compared to *miR172-abd* triple mutants under both LDs and SDs (Fig 5E–5G, Table 1, S1 Table). Although inactivation of *AP2* in the *mir172abd* mutant produced earlier flowering plants, the quadruple mutant bore more leaves and bolted later than Col-0 (Fig 5E and 5F, Table 1, S1 Table). These results demonstrate that ectopic and/or prolonged *AP2* activity contributes to the late-flowering phenotype of *mir172abd* mutants and that ectopic expression of other *AP2-LIKE* genes, including *TOE2* (Fig 5C and 5D), likely participates in repressing the floral transition of *mir172abd* mutants under LDs and SDs.

## Synergistic functions of *MIR172* and *FUL* during floral transition

In the mature inflorescence, the MADS-box TF FUL binds to the regulatory regions of *AP2*, *SNZ*, and *TOE1* to repress their transcription [47], whereas miR172 represses these genes at the posttranscriptional level. Whether this dual mechanism of repression of *AP2-LIKE* gene expression occurs during floral transition is unclear. Therefore, we tested the effect of combining *ful-2* and *mir172* mutations on flowering time (Fig 6A, S12A and S12B Fig).

We constructed the *ful-2 mir172a-2 b-3 c-1 d-3* quintuple mutant and compared its flowering time under LDs with Col-0 and parental controls (Fig 6A, Table 1, S1 Table). As previously observed, *ful-2* had a higher cauline leaf number (CLN) and the days to the first flower opening (DTF) was higher than Col-0 under these conditions, although the TLN of *ful-2* and Col-0 was similar (Fig 6A, Table 1, S1 Table). The quintuple *ful-2 mir172a-2 b-3 c-1 d-3* mutants bore more leaves and opened the first flower later than *mir172a-2 b-3 c-1 d-3* (Fig 6A, Table 1, S1 Table). These flowering time data indicated a synergistic genetic interaction when compared to the expected additive interaction between *ful-2* and *mir172a-2 b-3 c-1 d-3* [$\Delta$18.08 TLN ($p < 10^{-5}$), $\Delta$3.45 DTF ($p = 0.002$)].

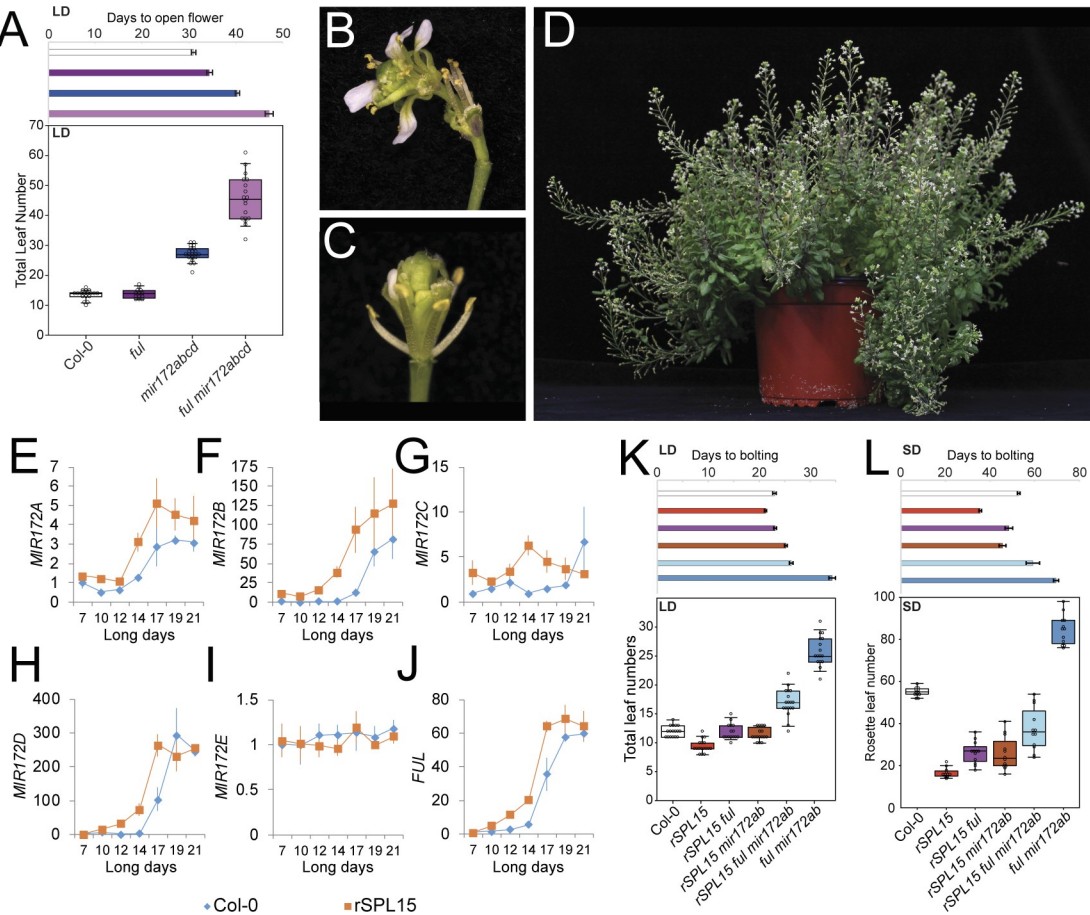

**Fig 6. Interplay between *Venus-rSPL15*, *MIR172*, and *FUL*.** (A) Graphs depicting DTF and TLN between combinations of *ful-2* and *mir172a-2 b-3 c-1 d-3*. Data underlying panel A are provided in S2 Data. (B, D) Photographs of (B, C) flowers arising from *ful-2 mir172a-2 b-3 c-1 d-3* plants and (D) an approximately 4-month-old *ful-2 mir172a-2 b-3 c-1 d-3* plant. (E–J) RT-qPCR results of (E) *MIR172A*, (F) *MIR172B*, (G) *MIR172C*, (H) *MIR172D*, (I) *MIR172E*, and (J) *FUL* expression in materials derived from the shoot apex of LD-grown Col-0 and *rSPL15-Venus* plants that were harvested at the indicated times after germination. Expression was normalized to 7 d Col-0 samples. Error bars indicate SEM of 3 independent biological replicates. Data from Col-0 is the same as in Fig 1, but tissue collection was originally paired with *rSPL15-Venus*. (K, L) Graphs depicting DTB and T/RLN between the indicated combinations of *rSPL15-Venus*, *mir172a-2*, *miR172b-3*, and *ful-2* mutants grown in (K) LD and (L) SD conditions. Data underlying panels (K) and (L) are provided in S2 Data. DTF, days to the first flower opening; FUL, FRUITFULL; LD, long-day; RLN, Rosette leaf number; RT-qPCR, quantitative reverse transcription PCR; SD, short-day; SEM, standard error of the mean; TLN, total leaf number.

Under SD conditions, *ful-2* was later flowering than Col-0 with slightly fewer rosette leaves, but more cauline leaves (S12B Fig, Table 1, S1 Table), in agreement with previous reports [58]. Under these conditions, the *ful-2 mir172a-2* and *ful-2 mir172b-3* double mutants were scored due to the strong late-flowering phenotype of the *ful-2 mir172a-2 b-3 c-1 d-3* quintuple mutant in LDs (Fig 6A). The *ful-2 mir172b-3* plants flowered substantially later than *ful-2 mir172a-2* or the single mutant progenitors under SDs (S12B Fig, Table 1, S1 Table), which supports a synergistic interaction between *FUL* and *MIR172B* under these conditions. Notably, under LDs, the relative severity of the double mutant flowering time was reversed compared to SDs, so that the *ful-2 mir172a-2* double mutant flowered later than *ful-2 mir172b-3* with similar leaf numbers (S12A Fig, Table 1, S1 Table). This result suggests a photoperiod-dependent input into different *MIR172* genes. Taken together, these data support a redundant role for FUL and miR172 in controlling flowering time. This genetic redundancy probably involves the

suppression of *AP2-LIKE* activity through transcriptional and posttranscriptional means, as was observed in mature inflorescences [47].

In further support of redundancy between FUL and mir172 in regulating *AP2-LIKE* genes in different developmental contexts, flowers of *ful-2 mir172a-2 b-3 c-1 d-3* plants lost meristem determinacy to a greater degree than *mir172* quintuple mutants (Fig 6B and 6C, S7A Fig), while *ful* mutants do not develop indeterminate flowers [47,48]. Furthermore, the longevity of *ful-2 mir172a-2 b-3 c-1 d-3* was extended far beyond *mir172a-2 b-3 c-1 d-3* or *ful-2*, with plants continuing to develop flowers after 5 months of growth under LDs (Fig 6D). These phenotypes are similar to those described for plants ectopically expressing *miR172*-resistant mRNAs of AP2-LIKE TFs, which delay flowering, increase inflorescence meristem activity, and reduce floral meristem determinacy [7,8,24,27,47].

### *MIR172* and *FUL* contribute to *SPL15* function during floral transition

SPL15 is an important regulator of flowering time under SDs, although it also plays a minor role under LD conditions that is circumvented by the photoperiod pathway [35,41]. Notably, miR172 and the mRNA of *FUL* are undetectable by *in situ* hybridization in the shoot apices of *spl15* mutants grown under SDs for 9 weeks, whereas in WT plants, both *FUL* mRNA and miR172 are detectable from 5 weeks [35]. Furthermore, SPL15 binds to the regulatory regions of *MIR172B* and *FUL*, indicating that these genes are direct targets of SPL15 [35]. To test further the idea that SPL15 activates transcription of *MIR172* genes and *FUL*, we tested the levels of *MIR172A-E* RNAs and *FUL* mRNA in a background containing an *SPL15* transgene controlled by its endogenous regulatory regions and engineered to be resistant to miR156 (*Venus-rSPL15*) (Fig 6E–6J) [35]. *FUL* and all *MIR172* genes, except *MIR172E*, were expressed at higher levels than in Col-0. We then combined *Venus-rSPL15*, *mir172a-2 b-3*, and *ful-2* and tested their flowering behaviors under both long and SDs (Fig 6K and 6L). Control experiments using *Venus-rSPL15* and Col-0 plants showed that *Venus-rSPL15* bolted earlier after germination and produced fewer leaves than Col-0 (Fig 6K and 6L, Table 1, S1 Table). However, the time between germination and the DTF was similar for both genotypes (S12C Fig). Thus, the time between DTB and DTF was extended in the *Venus-rSPL15* genotype relative to Col-0, which was also reflected in the longer stem length of *Venus-rSPL15* compared to Col-0 at the time DTF (S12C Fig). The introgression of either *ful-2* or *mir172a-2 b-3* into *Venus-rSPL15* delayed flowering time. This delay was enhanced further in a *Venus-rSPL15 ful-2 mir172a-2 b-3* background (Fig 6K and 6L, Table 1, S1 Table). However, *Venus-rSPL15 ful-2 mir172a-2 b-3* still flowered earlier than the *ful-2 mir172a-2 b-3* triple mutant. These data provide genetic support for *MIR172AB* and *FUL* acting downstream of SPL15, although SPL15 must also activate independent factors that can promote flowering in the absence of *MIR172AB* and *FUL* activity.

### Late flowering of *spl15* mutant is strongly enhanced by *mir172* or *ful* mutations

To determine whether *MIR172* and *FUL* only act downstream of SPL15 or whether they also have independent functions during floral transition, we constructed genotypes carrying the loss-of-function *spl15-1* mutation as well as *ful-2* or *mir172a-2* and *mir172b-3*, and measured their flowering times (Fig 7A, S12D and S12E Fig).

Under LD conditions, the flowering times of *spl15-1 mir172a-2 b-3* and *spl15-1 ful-2* were delayed relative to Col-0 and the parental genotypes (S12D and S12E Fig). The interaction between *spl15-1* and *mir172a-2 b-3* appeared to be largely additive when compared with the expected additive interaction [$\Delta$3.16 TLN ($p = 0.13$), $\Delta$DTF ($p = 0.94$)] (S12D Fig). This result

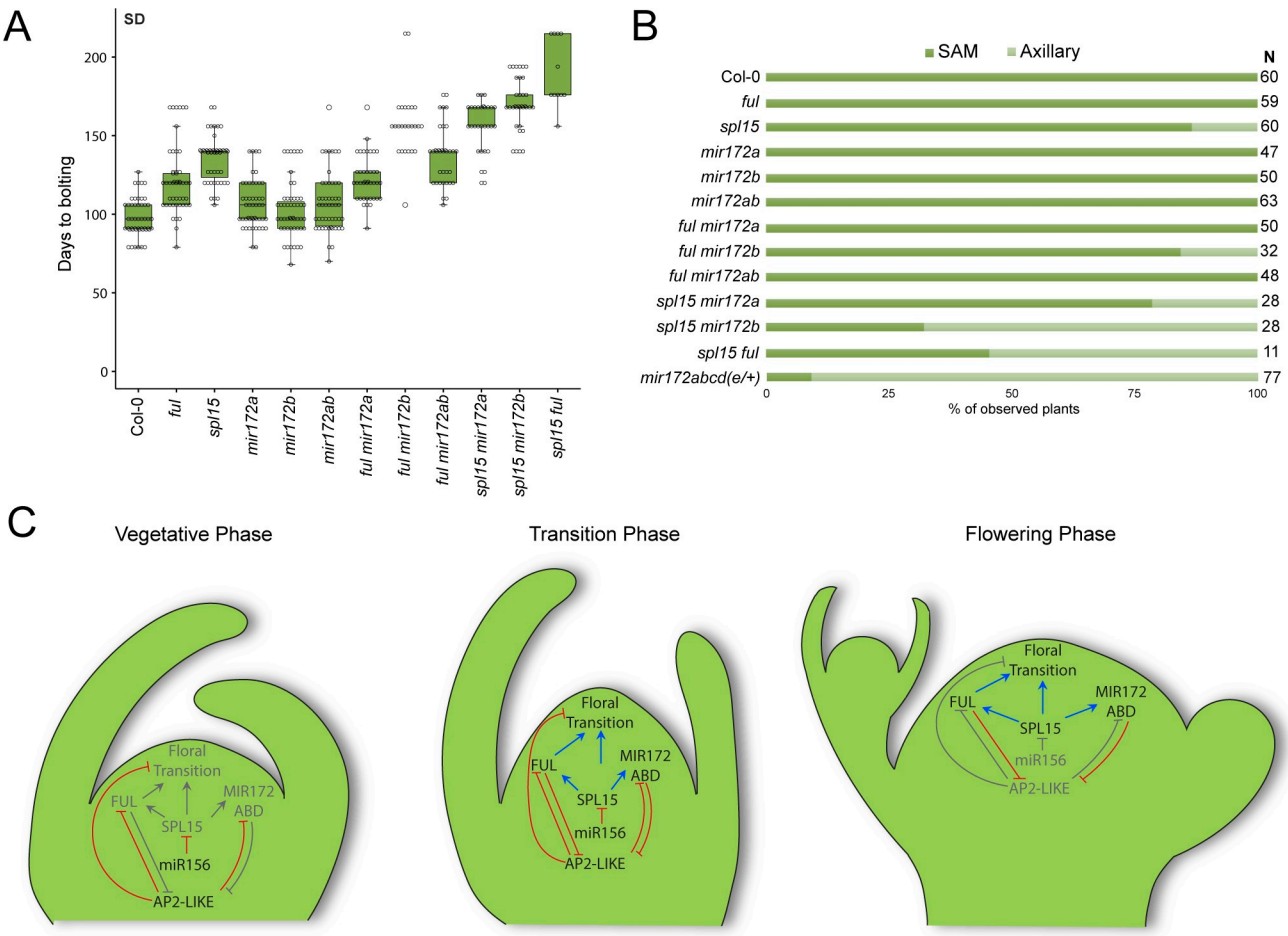

**Fig 7. Genetic interactions between *mir172*, *ful*, and *spl15* mutations.** (A) A box plot depicting DTB for plants carrying the indicated combinations of *spl15-1*, *ful-2*, and *mir172* mutations and grown in SD conditions. Data underlying panel A are provided in S2 Data. (B) A bar chart depicting the proportion of plants from (A) that flowered from the SAM or axillary meristems. The data for combinations between *ful-2*, *mir172a-2*, and *mir172b-3* are supplemented by data from S12B Fig. Data for *mir172abcd(e/+)* are taken from Fig 3B and are shown here for comparison. The total number, *N*, of plants analyzed is indicated. Data underlying panel B are provided in S2 Data. (C) Schematic to summarize the interactions coordinated by SPL15 during the floral transition. Red lines with flat caps indicate negative regulation, and blue arrows indicate positive regulation. Gray text and line color indicate inactivity at that stage of development. Black text indicates activity at that stage of development. AP2-LIKE, APETALA2-LIKE; DTB, days to bolting; FUL, FRUITFULL; SAM, shoot apical meristem; SD, short-day; SPL, SQUAMOSA PROMOTER BINDING PROTEIN-LIKE.

is consistent with *MIR172A* and *MIR172B* retaining activity in LD-grown *spl15-1* mutants, and therefore that *AP2-LIKE* genes are down-regulated at least partially independently of SPL15 under these conditions [35,41]. The interaction between *spl15-1* and *ful-2* appeared to be synergistic when compared to the expected additive interaction [Δ7.73 TLN (*p* = 0.0001), Δ3.05 DTF (*p* = 0.0001)] (S12E Fig). Given that *FUL* is required for complete FT activity [49,59], the loss of *FUL* function in LDs may enhance the importance of *SPL15* when the photoperiod pathway is partially disrupted, which indicates that *FUL* and *SPL15* do not act in a simple linear pathway under LDs.

Under SD conditions, many of the genotypes produced extremely high numbers of leaves, so we recorded only the number of days from sowing to bolting for many genotypes in a large-scale greenhouse experiment (Fig 7A), although we also performed smaller-scale experiments with combinations of *ful-2*, *mir172a-2*, and *mir172b-3* in growth cabinets, as described above (S12B Fig). Each of the parental genotypes behaved as previously observed as *spl15-1* bolted

much later than Col-0, *mir172a-2* bolted slightly later than Col-0, and *mir172b-3* bolted at a similar time to Col-0 (Fig 7A, Table 1, S1 Table). The *spl15-1 mir172a-2* double mutant bolted much later than *spl15-1* and *spl15-1 mir172b-3* bolted even later, with both interactions appearing to be synergistic compared to the expected additive interaction [Δ22.73 DTB; *p* = 0.0002] and [Δ35.48 DTB; *p* < 10–5], respectively (Fig 7A, Table 1, S1 Table). This result indicated that *MIR172A* and *MIR172B* activities are retained in SD-grown *spl15-1* mutants and promote flowering, although miR172 abundance is reduced in *spl15-1* [35].

Under SDs, *ful-2* also bolted later than Col-0, as expected (Fig 7A, Table 1, S1 Table). The *spl15-1 ful-2* double mutant bolted extremely late compared to *spl15-1* in a clearly synergistic interaction compared to the expected additive interaction (Δ33.78 DTB; *p* < 10–5) (Fig 7A, Table 1, S1 Table). This experiment also indicated that *FUL* activity is retained in *spl15-1*, although its expression is reduced and delayed [35]. Taken together, these data indicate that although *MIR172* and *FUL* are transcriptionally activated by SPL15, they retain activity in *spl15-1* mutants, and mutations in *MIR172* genes or *FUL* strongly enhance the *spl15* mutant phenotype.

## Flowering of primary shoots is more dependent on *SPL15*, *FUL*, and *MIR172* than flowering of axillary shoots

Several genotypes analyzed under SDs were severely impaired in floral transition at the SAM and initiated flowering from axillary branches, although most of these plants did eventually flower from the SAM as well; in contrast, WT Col-0 always flowered first from the SAM (Fig 7B). Under SDs, 14% (2/14) *spl15-1* and 14% (2/14) *ful-2 mir172b-3* plants initiated flowering from axillary meristems at the same time or before the SAM (S13A and S13B Fig). However, all *ful-2 mir172a-2* (14/14) or *ful-2 mir172a-2 b-3* (14/14) plants flowered first from the SAM. In an independent experiment, 13% (6/46) of *spl15-1* and 20% (3/15) of *ful-2 mir172b-3* plants bolted from axillary meristems at the same time or before the SAM. Once again, all *ful-2 mir172a-2* (36/36) and *ful-2 mir172a-2 b-3* (34/34) plants flowered from the SAM before the axillary meristems, as observed with WT Col-0 plants. Thus, in this respect, the *ful-2 mir172b-3* double mutant phenocopied *spl15-1*. Furthermore, the proportion of plants that bolted from axillary shoots first was higher in *spl15-1 mir172a-2* (21%, 6/28), *spl15-1 mir172b-3* (68%, 19/28), and *spl15-1 ful-2* (54%, 6/11) compared to *spl15-1* alone. This phenotype appears to be driven by an increase in AP2-LIKE activity as determined by the high proportion of *mir172* quadruple or quintuple mutants that flowered first from axillary meristems (91%, 70/77). These data from all experiments are summarized in Fig 7B. Furthermore, although this loss of apical dominance is more severe in short photoperiods, when these mutant combinations were grown in LD conditions, they exhibited similar architectural defects (S13D–S13H Fig). These results indicate that transcriptional activation of *FUL* and *MIR172* by SPL15 may be essential for flowering of the SAM in some plants and that these genes ensure the temporal progression of flowering from the shoot apex to axillary branches during shoot growth.

## Discussion

We used a combination of gene expression profiling and genome editing to demonstrate that miR172 is critical for flowering under noninductive SD photoperiods and that *MIR172ABD* contribute almost all of this activity. Under inductive LD photoperiods, *MIR172ABD* also promote flowering, but their contribution is hugely reduced compared to SDs, presumably because activation of the photoperiodic flowering pathway reduces the requirement for miR172. Delayed flowering of *mir172* mutants was partially conferred by extended expression of the miR172 target *AP2* at the shoot meristem. Previously, several *AP2-LIKE* genes were

shown to be directly negatively regulated at the transcriptional level by the FUL TF in the mature inflorescence meristem, and this was proposed to enhance the effect of miR172 on inflorescence meristem determinacy [47]. Our genetic data extend this interaction between miR172 and FUL to floral transition, as we find a synergistic interaction between *miR172* and *ful* mutations in delaying floral transition. We also use these genetic materials to extend the previous molecular analyses of Hyun and colleagues and show that the early flowering of a miR156-resistant version of *SPL15* is partially dependent on *FUL* and *MIR172* activities. Our data support a model whereby the promotion of both *FUL* and *MIR172* activities by SPL15 leads to the transcriptional and posttranscriptional suppression of *AP2-LIKE* activity, which is essential for floral transition at the SAM under SDs.

## Essential activity of *miR172* at the shoot apex

Flowering time analysis of *mir172* quintuple mutants combined with gene expression profiling showed that *MIR172A*, *MIR172B*, and *MIR172D* are the primary regulators of flowering time at the shoot apex (Figs 1 and 2). The product of *MIR172B* appears to also account for at least half of the total *miR172* pool in the leaves (Fig 4D), but other *MIR172* genes are also expressed in leaves (S8 Fig). Although miR172 increases *FT* expression indirectly, this regulation does not control the photoperiod response in a binary manner. Rather, it appears that miR172 modulates the response to photoperiod, perhaps as a means to provide developmental robustness, as has been observed for miRNAs in metazoans [60]. Such a role would explain the observation that *MIR172* expression appears to overlap with *AP2-LIKE* expression in the leaves and cotyledons rather than being mutually exclusive [7,22,57], although understanding their functions in leaves will require more thorough investigation at the tissue and cellular levels. Responses to other environmental stimuli, such as ambient temperature and winter cold [61], may be modulated by miR172 in a similar manner.

At the shoot apex, expression of *MIR172* and *AP2*, one of the 6 members of the *AP2-LIKE* family, appears to be mutually exclusive (Fig 5A). *AP2* mRNA and protein is rapidly cleared from the shoot apex during floral transition, which is probably important for a robust flowering response. This mechanism is particularly important under noninductive photoperiods as the photoperiod pathway appears to largely bypass the requirement for miR172 activity under inductive photoperiods. Genome-wide transcriptome profiling also showed that the mRNA of *TOE2* is highly increased under SDs in the *mir172abd* mutant background compared to Col-0, suggesting that TOE2 also contributes to late flowering of the mutant under these conditions. *TOE2* mRNA was previously shown to be reduced by overexpression of miR172, whereas the mRNA of *AP2* and other *AP2-L* genes was not [45]. It was proposed that these TFs feedback to repress their own transcription [21,45], so that reductions in their protein levels by overexpression of miR172 may lead to higher levels of their mRNAs. This feedback regulation together with cross-regulation among different members of the family makes it difficult to predict the temporal expression levels of AP2-L proteins in the *miR172* mutants. Indeed, we detected an unexpected delay in AP2-Venus accumulation in the apices of *mir172abd* mutants. Expression and genetic analysis of the whole AP2-L family will be required to fully decipher the interactions among these proteins.

Loss of *MIR172A-E* or *MIR172A*, *MIR172B*, and *MIR172D* delayed flowering under noninductive photoperiods by approximately 3 months compared to Col-0 controls. Furthermore, once these *mir172* mutants did flower, they mostly did not do so from the primary SAM but from axillary meristems, although usually the primary shoot did eventually flower. This loss of apical dominance suggests that *MIR172* activity is required to confer a coherent spatial response to inductive flowering cues and not only for timing the transition. Inactivation of

*AP2* partially suppressed the late flowering of *mir172abd* plants, although the *ap2 mir172abd* plants still flowered much later than Col-0 (Fig 5B and 5C). The effects of introducing mutations in other *AP2-L* genes, particularly *TOE2*, into *mir172abd* mutants remain to be tested. Our RNA-seq analysis suggests that several of these are expressed at the shoot apex and may also be important in controlling the floral transition at the shoot apex. Notably, Zhang and colleagues showed that *toe1* and *toe1 toe2* plants are extremely early flowering under both SD and LD conditions.

## Transcriptional and posttranscriptional regulation of *AP2-LIKE* activity by *FUL* and *miR172*

Previous work showed that FUL binds to regulatory regions in the proximity of *AP2-LIKE* genes (*AP2*, *TOE1*, *SNZ*, and *TOE3*) and that these genes are expressed at higher levels in *ful* mutant inflorescences [47]. Whether these interactions were relevant during floral transition was previously unclear. We tested genetic interactions between *ful* and *mir172* mutants under both LD and SD conditions (Figs 6 and 7). Under LDs, *ful mir172abcd* mutant plants flowered much later than either *mir172abcd* or *ful* plants. Furthermore, *ful mir172abcd* flowers exhibited a severe loss of determinacy, which did not occur in either parental line. Similarly, the longevity of *ful mir172abcd* plants was dramatically extended, with LD-grown plants continuing to produce flowers 4 months after germinating. This longevity is consistent with reports that the *AP2*-miR172 module is involved in global proliferative arrest [47]; however, it should be noted that the fertility of *ful mir172abcd* plants was extremely low. These phenotypes are closely related to those of plants in which *AP2-LIKE* genes are ectopically expressed [7,8,24,27,47]. Taken together, this genetic analysis indicates that FUL and miR172 control *AP2-LIKE* gene activity at both the transcriptional and posttranscriptional levels to regulate the floral transition, although this remains to be confirmed by examining expression of AP2 and AP2-L genes in the *ful mir172abd* background. Furthermore, as described above, analyzing transcript levels of these genes in different backgrounds is complicated by feedback regulation.

Expression analyses, including *in situ* hybridizations, indicate that miR172 acts prior to *FUL* at the shoot apex [35]. Therefore, we propose that miR172 is initially required to reduce *AP2-LIKE* activity and destabilize the vegetative state at the shoot meristem (Fig 7C). Furthermore, as AP2 directly represses *FUL* transcription [21], which is further supported by our analysis (S12F Fig), the depletion of AP2-LIKE TFs by miR172 is proposed to allow *FUL* transcription to increase. FUL would then bind to the promoters of AP2-LIKE genes initiating their transcriptional repression and thus enhancing the reduction of AP2-LIKE activity mediated by miR172 (Fig 7C). Affinity purification of FUL from inflorescences followed by protein mass spectrometry identified several chromatin remodeling proteins as interacting with FUL [62]. Therefore, binding of FUL to *AP2-LIKE* genes could mediate long-term transcriptional repression by compaction of the chromatin. We propose that these parallel mechanisms of posttranscriptional repression of AP2-LIKE gene expression by miR172 and transcriptional repression of the same genes by FUL provide a rapid and robust means to deplete these potent negative regulators of floral transition from the induced shoot apex as flowering proceeds (Fig 7C).

Expression of *FUL* and *MIR172* appears to be partially coordinated by SPL15 under both LDs and SDs, but its contribution is more important under SDs. When grown in SDs, *FUL* and miR172 are undetectable by RNA *in situ* hybridization in *spl15* mutants and SPL15 binds to the promoter regions of *FUL* and *MIR172B*. However, given that *mir172abd* mutants flower much later than *spl15-1* with *FUL* expression persisting in their SAMs (S12F Fig) and

inactivation of *FUL* or *MIR172A/B* can strongly enhance the phenotype of *spl15-1* (Fig 7A), miR172 and FUL activity must still occur in *spl15-1*. However, analysis of transgenic plants ectopically expressing a miR156-resistant version of SPL15 demonstrated that SPL15 can promote the transcription of *MIR172A*, *MIR172B*, *MIR172C*, and *MIR172D* (Fig 6E–6J). Furthermore, genetic analysis demonstrated that FUL and miR172 activity are partially required for rSPL15 activity, while combinations of *ful* and *mir172* mutants can partially phenocopy the *spl15* mutant (Figs 6K and 6L and 7A and 7B). These data support the notion that SPL15 acts through both *FUL* and *MIR172* to promote flowering, but that SPL15 also acts independently of these factors (Fig 7C). Similarly, other unidentified regulators must promote *FUL* and *MIR172* expression in parallel with SPL15. Thus, SPL15 mediates the transition from the stable vegetative state to the reproductive state by activating at least 2 repressors of AP2-LIKE TFs that act partially redundantly at different levels of regulation (Fig 7C). This is a form of feedforward circuit that has been described in other systems to mediate robust and stable repression of miRNA targets [60,63] and here likely plays a role in stabilizing the floral state.

## Materials and methods

### Plant growth and strains used

Plants were grown on soil in LD (16 h light/8 h dark) or SD (8 h light/16 h dark) conditions at 18 to 22˚C with a light intensity between 150 and 160 µmol m−2 s−1. All genotypes were in the *A. thaliana* (L.) ecotype Columbia-0 (Col-0) background. Previously published *A. thaliana* strains include *ap2-12* [21], *ful-2* [50], *35Spro:MIM172* [5], *spl15-1* [38], and *Venus-rSPL15* [35]. The following strains were generated in this study and are described below: *MIR172-A-NVG*, *MIR172B-NVG*, *MIR172C-NVG*, *MIR172D-NVG*, *MIR172E-NVG*, *mir172a-2*, *mir172b-3*, *mir172c-1*, *mir172d-2*, *mir172d-3*, *mir172e-1*, *mir172a-2 b-3*, *mir172a-2 b-3 c-1*, *mir172a-2 b-3 c-1 d-3*, *mir172a-2 b-3 c-1 d-3 e-1*, *mir172a-2 b-3 d-2*, *mir172a-2 b-3 d-2 ap2-12*, *spl15-1 ful-2*, *spl15-1 mir172a-2 b-3*, *ful-2 mir172a-2*, *ful-2 mir172b-3*, *ful-2 mir172a-2 b-3*, Venus-rSPL15 *ful-2 mir172a-2 b-3*, Venus-rSPL15 *ful-2*, Venus-rSPL15 *mir172a-2 b-3*, AP2_genomic-Venus, *ful-2 mir172a-2 b-3 c-1 d-3*; first generation plants harboring *UBQ10-pro*:CTRL, *UBQ10pro:MIR172A*, *UBQ10pro:mir172a-1*, *UBQ10pro:mir172a-2*, *UBQ10pro*:*MIR172B*, *UBQ10pro:mir172b-1*, *UBQ10pro:mir172b-3*, *UBQ10pro:MIR172D*, *UBQ10pro*:*mir172d-2*, *UBQ10pro:mir172d-3*, *UBQ10pro:MIR172E* and *UBQ10pro:mir172e-1*.

### Cloning MIR172-NVG reporter constructs

Polymerase Incomplete Primer Extension (PIPE) [64] was used to generate the NVG-stb205 and MIR172(A-E)-NVG-stb205 plasmids. The NVG-stb205 plasmid, which is a variant of pDONR201, was generated with primers oDM-462-465 (S2 Table). To generate the MIR172 (A-E)-NVG-stb205 plasmids, initially, the entire intergenic region containing the *MIR172* coding region of each gene was placed in stb205 with primers oDM-11-20 (S2 Table). The NVG coding region was used to replace the miRNA hairpin coding sequence in each *MIR172* gene using primers oDM-31-50 (S2 Table). Either Phusion (New England Biolabs, United States of America) or PrimeSTAR GXL (TaKaRa Bio, Japan) high-fidelity polymerases were used for DNA amplification. The PCR products derived from plasmid templates were treated with *Dpn*I (New England Biolabs) overnight at 37˚C to remove the template DNA. After purification of each PCR product with the NucleoSpin PCR clean-up kit (Macherey Nagel, Germany), a molar ratio of 1:10 plasmid:insert PCR products were mixed and 100-µL chemically competent Dh5α *Escherichia coli* were transformed using the heat shock method. Transformed bacteria were incubated with 500-µL LB for 1 h at 37˚C in a shaker. The entire liquid culture was then centrifuged for 5 min at 5 krpm, and 500 µL of the supernatant was removed. The

other 100 µL of supernatant was used to resuspend the bacterial pellet, which was plated on LB-agar plates supplemented with 50 µg/mL kanamycin and incubated at 37˚C overnight. Colonies were tested by PCR using primers in S2 Table, and positive colonies were used to inoculate 5-mL liquid LB culture, which were supplemented with 50 µg/mL kanamycin and were grown overnight at 37˚C. Plasmids were extracted using the NucleoSpin Plasmid extraction kit (Macherey Nagel, Germany), and the success of the cloning was initially determined by DNA restriction enzyme analysis. Plasmids with the correct restriction patterns were subjected to Sanger sequencing (Eurofins Genomics, Germany) to verify that the correct sequences were present in the plasmids. MIR172(A-E)-NVG sequences were then transferred to the p*GW binary vector [65] using the Gateway LR clonase II enzyme mix (Invitrogen, USA), according to the manufacturer's protocol. After transformation protocols as described above, a 100-µL culture was plated on LB-agar plates supplemented with 100 µg/mL spectinomycin. Colonies were tested by PCR using primers in S2 Table. The success of the cloning was determined as described above although Sanger sequencing was not repeated. Plasmids were transformed into *Agrobacterium tumefaciens* strain GV3101 by electroporation, and 500 µL of LB was added and incubated at 28˚C for 3 to 4 h. Moreover, 100 µL of the culture was plated on LB-agar plates supplemented with 100 µg/mL spectinomycin, 100 µg/mL rifampicin, and 25 µg/mL gentamicin and incubated at 28˚C. Positive colonies were determined by colony PCR using primers in S2 Table, and cultures were grown in LB medium supplemented with 100 µg/mL spectinomycin, 25 µg/mL gentamicin, and 100 µg/mL carbenicillin.

## Cloning overexpression of wild-type and mutant *MIR172* alleles

The *A. thaliana UBIQUITIN10* promoter and the octopine synthase terminator sequences (from pBJ36 [66]) were introduced to the 5′ and 3′ sides of the multiple cloning site of pGREEN-0229 [67], respectively, using restriction enzyme based cloning to generate *pGREEN-0229-UBQ10pro-3′OCS* (S2 Table). The WT and mutant alleles for *MIR172* were amplified with the Phusion (New England Biolabs) high-fidelity polymerase and primers that contained *Xho*I and *Not*I sites (S2 Table). PCR products and pGREEN-0229-UBQ10-3′OCS were incubated with *Xho*I and *Not*I (New England Biolabs) overnight at 37˚C. After gel extraction and purification of the digested products with the NucleoSpin PCR clean-up kit (Macherey Nagel, Germany), they were incubated at 4˚C overnight with T4 DNA ligase (New England Biolabs). The ligation reaction was then transformed into Dh5α using the heat shock method. Transformed bacteria were incubated with 500 µL of LB for 1 h at 37˚C in a shaker. Moreover, 100 µL of the supernatant was plated on LB-agar plates supplemented with 50 µg/mL kanamycin and incubated at 37˚C overnight. Success of the cloning, transformation of *A. tumefaciens* strain GV3101 that contained the pSOUP helper plasmid, and culturing of positive colonies was performed as described above with the exception that 10 µg/mL of tetracycline was also added to the LB/LB-agar media.

## Single guide RNA cloning, T7E1 assays, and CRISPR genotyping

Single guide RNA (SgRNA) cloning was performed according to the methods in [52,53], and the appropriate primers are listed in S2 Table. Pure genomic DNA was extracted from the cauline leaves of T1 plants [68]. The region in which the Cas-9 protein was targeted was flanked by 2 sets of nested primers (S2 Table). Amplified products were reannealed by denaturing them at high temperature and slowly decreasing the temperature [53]. Moreover, 1 µL of the T7E1 enzyme (New England Biolabs) was then introduced to 10 µL of the PCR reaction supplemented with sterile distilled $H_2O$ and the appropriate buffer (NEB), and samples were incubated at 37˚C for 20 min in a thermocycler. Incubated products were then run on an EtBr-

supplemented 3% agarose gel and visualized under an ultraviolet lamp (S4 Fig). Seeds from plants that were positive in the T7E1 assay were then planted, and plants with stable mutations were identified through a combination of T7E1 assays, PCR amplification, and Sanger sequencing. Plants lacking the Cas-9 insertion were identified using a leaf-based phosphino-thricin (PPT) resistance assay (S4N Fig) and genotyping (S2 Table).

## Cloning of AP2_genomic-Venus reporter construct

PIPE [64] was used to generate the plasmids csAP2-pDONR201, pAP2-AP2-pDONR201, pAP2-AP2-Venus-pDONR201, and pAP2-AP2-Venus-pEarlyGate301. All PCR amplifications were done with Phusion Enzyme (New England Biolabs) following the manufacturer's recommendations. Amplification of the AP2 coding sequence was performed with Q236+Q272 and was cloned into pDONR201 by BP reaction to generate csAP2-pDONR201. Q273+fa20r were used to linearize csAP2-pDONR201, and Q274+Q275 were used to amplify the *AP2* promoter to generate pAP2-AP2-Venus-pDONR201. The Venus coding region was amplified using Q058+Q359, and Q357+Q358 were used to linearize pAP2-AP2-Venus-pDONR201 to generate pAP2-AP2-Venus-pDONR201. The pAP2-AP2-Venus sequence was then transferred to the pEarlyGate binary vector [65] using the Gateway LR clonase II enzyme mix (Invitrogen), according to the manufacturer's protocol. Transformation of bacteria was performed as described above.

## Plant transformation and selection

The floral dip method [69] was used to transform *A. thaliana*. A solution containing the appropriate *A. tumefaciens* strain, 10% w/v sucrose, and 0.025% v/v Silwett L-77 was used to dip the flowers of *A. thaliana*. Plants were then incubated in the dark at room temperature overnight before being placed in LD-grown conditions. For selection of plants transformed with a bar selection cassette, seeds were germinated on soil, and seedlings were sprayed with 200 μg/ml ammonium-glufosinate. For selection of plants transformed with a hygromycin-resistance selection cassette, sterilized seeds were plated on MS agar supplemented with 12 μg/ml hygromycin, covered, and placed in a growth chamber for 3 d. After 3 d, the cover was removed, and seedlings were grown until a noticeable difference in hypocotyl length was observed. Seedlings with longer hypocotyls were presumed to be resistant to hygromycin. Putative transformants were genotyped using appropriate primers for confirmation.

## Phosphinothricin (PPT) resistance assay

Personal communications with several researchers from independent groups highlighted the problem with segregating the Cas-9 containing T-DNAs from the mutation of interest, as they often appeared to be genetically linked. In order to identify plants lacking the Cas-9 containing T-DNAs without needing to genotype hundreds of plants or propagate seeds from hundreds of plants, we used a plate based PPT-resistance leaf assay. MS-agar plates (without sucrose) supplemented with 25 μg/mL PPT were poured. A leaf (preferably a young leaf but part of an old leaf is also suitable) was placed on the MS-agar and pressed down until it stuck to the agar. Plates were then sealed with medical tape and placed in a growth cabinet for 2 to 3 days. Leaves derived from PPT-resistant plants were dark green, whereas leaves from PPT-susceptible plants were light green (S4N Fig). Follow-up genotyping by PCR confirmed the absence or presence of the Cas-9 T-DNA (S2 Table). This assay was not performed in sterile conditions, and fungal growth was kept to a minimum by using sucrose-free MS-agar plates and limiting the time the MS-agar plates were open.

## ClearSee and confocal imaging

The ClearSee reagent (10% (w/v) xylitol, 15% (w/v) sodium deoxycholate, and 25% (w/v) urea) was prepared as previously described [70]. Tissue was dissected at approximately zeitgeber time (ZT) 6 and placed in 4˚C 4% paraformaldehyde prepared in PBS (pH 7), which was resting on ice. Tissue was then subjected to a vacuum of approximately 700 mbar at room temperature for 1 h. The vacuum was then released, and samples were incubated for a further 1 h. Samples were washed with 1x PBS 3 times and then incubated with ClearSee in the dark for up to 14 d. The shoot meristems were visualized with a LSM780 (Zeiss, Germany) laser scanning confocal microscope, and the settings were optimized for the visualization of the Venus fluorescent protein (laser wavelength, 514 nm; detection wavelength, 517 to 569 nm) and the SCRI Renaissance 2200 dye (laser wavelength, 405 nm; detection wavelength, 410 to 503 nm), which stains plant cell walls [71]. The same settings were used for each sample; however, the gain for visualizing the plant cell walls by Renaissance staining was modified for presentation purposes. This does not change the interpretation of the results, and original images are available upon request. The Renaissance dye was added to the samples 12 to 24 h before imaging. For Fig 5A, the samples in a Col-0 and *mir172abd* background are unpaired. The Col-0 background samples were imaged with a lower 514-nm gain. A paired Col-0 background sample was performed, but the signal was dimmer probably because of the quenching effects of ClearSee; however, the data were similar.

## RNA extraction and RT-qPCR

Plant tissue was harvested at the indicated times on liquid nitrogen and immediately frozen at −80˚C. Samples were processed with the RNeasy Plant Mini Kit (Qiagen, the Netherlands) according to the manufacturer's protocol. RNA was quantified using a NanoDrop 1000 spectrophotometer (Thermo Fisher Scientific, USA), and up to 5 μg of RNA was incubated with DNase I (TURBO DNA-free kit, Invitrogen) to remove genomic DNA according to the manufacturer's protocol. After quantification, 1 μg of DNA-free RNA was used in a reverse transcription reaction (Superscript IV first-strand synthesis system, Invitrogen) and diluted 1:5 with pure and sterile $H_2O$. Moreover, 1 μL of the diluted reverse transcription reaction was used in a PCR reaction with iQ™ SYBR green supermix (Bio-Rad, USA). Relative transcript abundance was determined with a Roche LightCycler 480 system. Measurements were taken for 3 independent biological triplicates, and the average transcript abundance for each biological sample was determined by performing 2 to 3 technical repeats. LightCycler melting curves were obtained for the reactions, revealing single peak melting curves for most amplification products. The amplification data were analyzed using the second derivative maximum method, and resulting Cp values were converted into relative expression values using the comparative cycle threshold method [72]. One reference gene (REF1 (*PROTEIN PHOSPHATASE 2A* (*PP2A*)) was used to normalize the data.

## DNA extraction and genotyping

Pure genomic DNA was extracted for T7E1 assays according to [68], for next generation sequencing with the DNeasy plant mini kit (Qiagen) according to manufacturer's instructions, and for genotyping according to [73]. Platinum Taq DNA polymerase (Invitrogen) was used for all PCR assays except for cloning DNA fragments.

## Statistical analysis

We have implemented some recommendations from *The American Statistician* [74]. Specifically, we avoid the term "statistically significant," and we report *p*-values as continuous where

possible. We have supplied a Supporting information table in which all *p*-values are reported in addition to a description of the dataset (S1 Table). Although grouping data based on statistical thresholds is not recommended [74], we have provided a grouping threshold in Table 1 and S1 Table as we believe that it can be a helpful guide, but we do not interpret our results on the basis of these groupings.

Data were analyzed using the *R studio* software [75]. The distribution and variance of the data were analyzed initially by visualization and the base functions *shapiro.test*, and *bartlett.test* and/or *fligner.test*, respectively. Data were analyzed using 1-way analysis of variance (ANOVA) if data were deemed to be normally distributed (according to a combination of statistical analysis and data visualization) and if the variance of the data were equal ($p > 0.01$ for at least one of *bartlett.test* or *fligner.test*). If these assumptions were violated, data were transformed to meet these assumptions. The Kruskal–Wallis test was used (*kruskal.test*) if the data were not normally distributed; however, $p > 0.01$ was required for at least one of *bartlett.test* or *fligner.test*. If $p < 0.01$ for *bartlett.test* and *fligner.test*, and data transformation did not off-set the problem, the Brown–Forsythe (*bf.test* from the package *onewaytests*) test was implemented [76].

Post hoc tests were employed if a significant result was obtained from the above tests ($p < 0.01$). The base function *pairwise.t.test* was used following an ANOVA test, with analysis being paired or unpaired depending on the experimental design. Dunn test (*dunn.test* [77]) was used following the Kruskal–Wallis test. Games Howell test was used following the Brown–Forsythe test from the package *userfriendlyscience* [78]. Adjustments for multiple corrections were performed in each case using the Benjamini–Hochberg method [79].

Tests for genetic synergism were done by subtracting observed values from the mean of the reference Col-0. Then means and variances were added to simulate the expected additive effect of the 2 alleles. This was compared with the observed differences between the higher-order mutant and Col-0 using an unpaired 2-tailed Welch *t* test.

For the box plots, the box encapsulates the 25th to 75th percentile, and the error bars encapsulate the 10th and 90th percentiles. The horizontal line running through the box indicates the median, and each point represents an individual plant. Bars represent the mean value, and the error bars represent the standard error of the mean (SEM). Graphs were modified for style using Adobe Illustrator.

## GUS staining

GUS staining was performed according to [80] with some modifications. Briefly, samples were harvested at approximately ZT6 in ice-cold 90% acetone and vacuum infiltrated at approximately 700 mbar for 10 min at room temperature. The vacuum was then released, and samples were incubated for a further 20 min. The acetone was removed and replaced with cold staining buffer (50 mM $PO_4$, 0.2% Triton X-100, 2 mM ferro-ferricyanide) without X- glucuronidase (X-gluc) and vacuum infiltrated at approximately 700 mbar on ice for 10 min. Cold staining solution without X-gluc was removed, and staining solution with 2 mM X-gluc was added, and samples were vacuum infiltrated on ice for up to 20 min with the vacuum being released and reapplied 2 to 3 times. Samples were then incubated at 37˚C for 2 h or overnight. After this incubation, samples were submerged in 70% ethanol, which was refreshed until the chlorophyll was removed. Images were taken with a Zeiss SMZ-25 (Zeiss) stereomicroscope or a digital SLR camera.

## RNA *in situ* hybridization

Plants were grown in LD conditions for the specified time. *In situ* hybridization was performed as previously described [59]. The osa-miR172a miRCURY LNA miRNA detection probe

(Qiagen, Cat No. YD00618724) was used to detect *miR172* RNA. A probe for *AP2* was synthesized as described in [31].

## Whole genome sequencing and off-target analysis

Library preparations of genomic DNA derived from quintuple *mir172abcde* mutant plants were performed using the NEBNext Ultra II FS DNA library preparation kit for Illumina. Whole genome sequencing (20X coverage) was performed with an Illumina HiSeq3000 with single-end 150 bp reads. Potential off-targets were identified by comparing the sgRNA sequences with the *A. thaliana* genome, allowing for up to 5 mismatches using the dedicated off-target search program *CAS-OFFinder* [81]. In order to detect mutations, the raw single-end reads were processed by removing the adapter sequences using *cutadapt* [82] and then trimming low-quality ends using *Trimmomatic* [83]. The cleaned reads were mapped to the *A. thaliana* TAIR10 genome using *BWA* [84] with default settings. *BWA* outputs both primary, and if found, secondary alignments. In this study, we only retained primary alignments with a mapping quality of at least 30. *Samtools mpileup* [85] (default settings) was used in conjunction with *bcftools* [86] for initial SNP calling. SNPs were filtered by minimum quality of 25, minimum depth of 5. Additionally, all regions of the genome that were devoid of any mapped read were labeled as potential larger deletions. Finally, we searched for overlap between the predicted off-targets, the SNPs, and larger deletions.

## RNA secondary structure analysis

Sequences from 24 nucleotides downstream and upstream of the miR172/miR172* duplex coding sequences of WT and mutant *MIR172* alleles were processed by mFold version 2.3 [87] with standard settings except that the temperature was set at 22˚C. The predicted structure with the lowest minimum free energy is presented.

## RNA extraction and RNA sequencing transcript profiling

The shoot apex of Col-0 and *mir172abd* were dissected under a stereo microscope at 10, 12, 14, and 17 days in LD conditions and 3, 4, 5, and 6 weeks in SD conditions in 3 independent biological replicates. Total RNA was extracted using the RNeasy Plant Mini Kit (Qiagen, USA) and subjected to DNase treatment using the TURBO DNase (Invitrogen). Poly(A) RNA enrichment, library preparation, and sequencing were carried out at the MPIPZ Genome Center, Cologne, Germany using the following conditions: The RNAs were processed by poly-A enrichment followed by application of basic components of "NEBNext Ultra II Directional RNA Library Prep Kit for Illumina" with a homebrew barcoding regime. Sequencing was performed on a HiSeq3000 sequencer by sequencing-by- synthesis with $1 \times 150$ bp single-read length. Sequence reads were preprocessed to remove any residual adaptors with CutAdapt, and the low-quality bases (Q<15) were trimmed from the ends with Trimmomatic [82,83]. Only reads with a minimum length of 50 nucleotides were kept. Salmon was used to quantify the abundance of transcripts from the *Arabidopsis* reference genome Reference Transcript Dataset for *Arabidopsis* (including guanine/cytosine bias, unstranded samples) [88,89]. Fragments Per Kilobase of transcript per Million (FPKM) values and corrected *p*-values were obtained using DESeq2 by comparing Col-0 to *mir172abd* in each time point using standard settings. Data from this RNA-seq transcript profiling are deposited in the BioProject database (https://www.ncbi.nlm.nih.gov/bioproject) under the accession number PRJNA669254.

## Gene accession numbers

Annotations and sequence data of the genes studied in this article can be found at The *Arabidopsis* Information Resource (TAIR) under the following accession numbers: *MIR172A* (*At2g28056*), *MIR172B* (*At5g04275*), *MIR172C* (*At3g11435*), *MIR172D* (*At3g55512*), *MIR172E* (*At5g59505*), *AP2* (*At4g39620*), *TOE1* (At2g28550), *TOE2* (*At5g60120*), *TOE3* (*At5g67180*), *SNZ* (*At2g39250*), *SMZ* (*At3g54990*), *FUL* (*At5g60910*), *SPL15* (*At3g57920*), *FT* (*At1g65480*), and *AP1* (*At1g69120*).

## Supporting information

**S1 Fig. Schematics of the reporter constructs generated for *MIR172A-E*.** The NVG coding sequence is depicted by a short bright red box (NLS), a yellow box (Venus), and a blue box (GUS). This coding sequence was used to replace the hairpin region (dark red and orange lines) in each *MIR172* gene. The beige boxes represent exons, gray lines represent introns, and black lines represent intergenic sequences including untranslated regions. Brown boxes indicate adjacent genes. *At3g55513* is annotated as encoding a hypothetical protein, whereas *At3g08495* and *At2g08215* are annotated as expressing long noncoding RNAs, respectively, and were therefore included in the reporter constructs for *MIR172A* and *MIR172D* as they may contain regulatory elements relevant to their expression. The complete genomic structure has only been determined for *MIR172A* and *MIR172B*. Therefore, the most upstream transcription start site for *MIR172C* and *MIR172E* described in [90] were used. RNA-seq data available from Araport [91] were used to estimate which regions surrounding the hairpin coding regions of *MIR172A-C* are transcribed. Black arrows beneath the schematics indicate ATGs upstream of the NVG reporter, whereas gray arrows indicate ATG that were present before the *MIR172* hairpin coding region that were omitted from the reporter constructs. ATG, adenine, thymine, guanine; GUS, β-glucuronidase enzyme; NVG, NLS-Venus-GUS; RNA-seq, RNA-sequencing.
(TIF)

**S2 Fig. Expression patterns of *MIR172-NVG* transgenic reporter lines in shoot apices grown in LDs.** (A–F) Confocal laser scanning micrographs of the shoot apices of additional independent homozygous transgenic lines of (A) *MIR172A-NVG*, (B-C) *MIR172B-NVG*, (D, E) *MIR172C-NVG*, and (F) *MIR172D-NVG* transgenic plants grown in LD conditions and harvested at the indicated times after germination. (B, C) Note the presence of fluorescence in the L1 of each independent *MIR172B-NVG* line (inset, arrowheads). Fluorescence from the Venus protein is artificially colored in green, and the fluorescence from the Renaissance dye is artificially colored in magenta. LD, long-day; NVG, NLS-Venus-GUS.
(TIF)

**S3 Fig. Expression patterns of second generation *MIR172-NVG* transgenic reporter lines in shoot apices grown under LD conditions.** Confocal laser scanning micrographs of the shoot apices of 3 independent (A–C) *MIR172A-NVG*, (D–F) *MIR172B-NVG*, (G–I) *MIR172C-NVG*, and 1 independent (J) *MIR172D-NVG* transgenic reporter line(s). (D) Note the presence of fluorescence in the L1 of *MIR172B-NVG*[#2] (inset, arrowheads). Fluorescence from the Venus protein is artificially colored in green, and the fluorescence from the Renaissance dye is artificially colored in magenta. LD, long-day; NVG, NLS-Venus-GUS.
(TIF)

**S4 Fig. Isolation of CRISPR-Cas-9-induced mutants of *MIR172A-E*.** (A–E) The positions of each sgRNA are indicated (blue indicates a functional sgRNA, red indicates a non-functional

sgRNA, according to the T7E1 assays). The miR172* and miR172 coding sequences are highlighted in red and orange, respectively. The PAM sequence required for functionality of the Cas-9-sgRNA complex is highlighted in purple. (F–H, J–L) Agarose gels of the indicated T7E1 assays of functional sgRNAs. DNA was derived from $T_1$ plants transformed with the CRISPR-Cas-9 system from [53]. The expected patterns of restriction digest are indicated below the photograph, and the associated sgRNA is indicated above. (I) An agarose gel of a PCR of *MIR172C* using DNA derived from $T_1$ plants harboring the CRISPR-Cas-9 system from [52], indicating the presence of a deletion (asterisk). Data underlying panels F to L are provided in S3 Data. (M) The sequences of WT *MIR172A-E* genes and the mutants identified by CRISPR-Cas-9. The miR172* and miR172 coding sequences are highlighted in red and orange, respectively, while the sequences required for the first processing cleavages by DCL1 are highlighted in green. Note that *mir172d-3* contains several SNPs (highlighted in red) rather than a deletion. (N) An example of a PPT-resistance assay to identify plants that lack the Cas-9-containing T-DNA. Young leaves were places on an MS-agar (-sucrose) plate supplemented with 25 μg/mL PPT and incubated in a growth chamber for 3 days. The leaves derived from plants lacking the Cas-9-containing T-DNA are light green (red circles), whereas the leaves derived from plants harboring the Cas-9-containing T-DNA are dark green. Cas-9, CRISPR associated protein-9; CLN, cauline leaf number; CRISPR, clustered regularly interspaced short palindromic repeats; PAM, protospacer adjacent motif; PPT, phosphinothricin; sgRNA, single guide RNA; WT, wild-type.
(TIF)

**S5 Fig. Predicted secondary structures of WT and mutant pri-miRNAs.** The predicted secondary structure of the pri-miRNAs derived from (A) *MIR172A*, (B) *mir172a-1*, (C) *mir172a-2*, (D) *MIR172B*, (E) mir172b-1, and (F) mir172b-3 are shown. The miR172* and miR172 coding sequences are highlighted in red and orange, respectively. miRNA, microRNA; WT, wild-type.
(TIF)

**S6 Fig. Expression patterns of *MIR172-NVG* transgenic reporter lines in shoot apices grown under SD conditions.** (A–F) Confocal laser scanning micrographs of the shoot apices of additional independent transgenic reporter lines of (A) *MIR172A-NVG*, (B, C) *MIR172B-NVG*, (D, E) *MIR172C-NVG*, and (F) *MIR172D-NVG* transgenic plants grown in LD conditions and harvested at the indicated times after germination. Fluorescence from the Venus protein is artificially colored in green, and the fluorescence from the Renaissance dye is artificially colored in magenta. LD, long-day; NVG, NLS-Venus-GUS; SD, short-day.
(TIF)

**S7 Fig. Leaf initiation rates and morphologies of *mir172* mutants the *35Spro:MIM172* line.** (A) A photograph of *mir172a-2 b-3 c-1 d-3* siliques grown in LD conditions from a single plant. The ages of the siliques are in order from left (early arising) to right (late arising). Note that the phenotype is most severe in the first flowers/siliques to arise. (B) A photograph of *mir172a-2 b-3 d-2* inflorescence. Note the extremely serrated cauline leaves (arrowheads). (C) Graphs depicting the TLN and DTB of Col-0, *mir172c-1*, *mir172e-1*, and *mir172c-1 e-1* grown under LD conditions. (D) The number of leaves present on SD-grown plants that were larger than 0.25 cm on each indicated day. Error bars indicate SEM. ($N$ = 12–16). Statistical comparisons relative to Col-0 at 40d: mir172a; Δ4.18 RLN; $p$ = 0.001), (mir172ab; Δ5.03 RLN; $p < 10^{-5}$), (mir172abc; Δ3.75 RLN; $p$ = 0.003), (mir172abcd(e/+); Δ4.69 RLN; $p$ = 0.0004). The *Venus-rSPL15* genotype was used here as a control. Data underlying panel D are provided in S2 Data. (E) Micrographs of RNA *in situ* hybridizations using a commercial probe designed to

recognize miR172 of Col-0 (upper panels) and *mir172a-2 b-3 c-1 d-3 e-1* (lower panels) apices grown under SD conditions for the indicated times. (F) A representative photograph of the morphologies of plants of the same age grown under SD conditions. The genotypes used in each experiment were Col-0, *mir172a-2*, *mir172a-2 b-3*, *mir172a-2 b-3 c-1*, *mir172a-2 b-3 c-1 d-3 e-1/+*, *35Spro:MIM172*, and *Venus-rSPL15*. DTB, days to bolting; LD, long-day; SEM, standard error of the mean; SD, short-day; TLN, total leaf number.
(TIF)

**S8 Fig. GUS staining of *MIR172-NVG* transgenic reporter lines grown in LD conditions.**
Photographs of 2h GUS staining of independent homozygous transformants for (A, B) *MIR172A-NVG*, (C, D) *MIR172B-NVG*, (E, F) *MIR172C-NVG*, and (G, H) *MIR172D-NVG* grown under LD conditions and harvested at the indicated times. Red arrowheads indicate the presence of weak GUS staining. GUS, β-glucuronidase enzyme; LD, long-day; NVG, NLS-Venus-GUS.
(TIF)

**S9 Fig. Prolonged GUS staining of second generation *MIR172-NVG* transgenic lines grown in LD conditions.** Photographs of overnight GUS staining of 3 independent second generation transformants for (A–C) *MIR172A-NVG*, (D, E) *MIR172B-NVG*, (G, I) *MIR172C-NVG*, (L–N) *MIR172D-NVG*, and (J, K) 2 independent second generation transformants grown under LD conditions and harvested 21 d after germination. (O) A graph indicating the log2 TP2M for each miR172 isoform in the leaves of *A. thaliana* (data from [56]). Note that the isoform arising from *MIR172A* and *MIR172B*, and *MIR172C* and *MIR172D*, are identical, respectively. Data underlying panel O are provided in S1 Data. GUS, β-glucuronidase enzyme; LD, long-day; NVG, NLS-Venus-GUS; TP2M, transcripts per 2 million.
(TIF)

**S10 Fig. Chromosomal locations of *MIR172, AP2-LIKE, SPL15*, and FUL genes.** Visualization of the chromosomal locations of each *MIR172, AP2-LIKE, SPL15*, and *FUL* genes in *A. thaliana*. Map was created with the Chromosome Map Tool provided by TAIR (www.arabidopsis.org). AP2-LIKE, APETALA2-LIKE; FUL, FRUITFULL; SPL, SQUAMOSA PROMOTER BINDING PROTEIN-LIKE; TAIR, The *Arabidopsis* Information Resource.
(TIF)

**S11 Fig. Schematic of the reporter construct generated for APETALA2 and RNA-seq data for *AP2-LIKE* gene transcript levels in Col and *mir172abd* mutants.** (A) The Venus coding sequence is depicted by a yellow box. This coding sequence was fused in frame to the final exon of *AP2* by removing the stop codon of *AP2*. The beige boxes represent exons, gray lines represent introns, and black lines represent intergenic sequences including untranslated regions. Brown boxes indicate adjacent genes. *At4g09445/At4g09415* are annotated as long noncoding RNAs, and *At4g09415* is annotated as "other RNA" and were therefore included in the reporter construct as they may contain regulatory elements important for AP2 expression. (B, C) Graphs depicting RNA-seq-derived data of the mRNA levels in apices of *SMZ, SNZ, TOE1, TOE3*, and *AP1* in Col-0 and *miR172abd* at the indicated time points after germination in (B) LDs and (C) SDs. Error bars represent the standard deviation of 3 biological replicates. Data underlying panels (B) and (C) are provided in S1 Data. AP2, APETALA2; AP2-LIKE, APETALA2-LIKE; LD, long-day; RNA-seq, RNA-sequencing; SD, short-day.
(TIF)

**S12 Fig. Interplay between *Venus-rSPL15, ful-2, mir172*, and *spl15* in long and short days.**
(A, B) Graphs depicting DTF and TLN between the indicated combinations of *ful-2* and

*mir172* mutants grown in (A) long and (B) SD conditions. (C) A bar chart depicting the DTB, DTF, the duration between DTB and DTF (ΔDTF-DTB), and the stem length at flowering. (D, E) Graphs depicting the days to open flower and TLN of combinations between *spl15-1*, *ful-2*, and *mir172-a2 b-3* in LD conditions. Data underlying panels A to E are provided in S2 Data. (F) Micrographs of RNA *in situ* hybridizations using a probe designed to recognize *FUL* of Col-0 (upper panels) and *mir172a-2 b-3 c-1 d-3 e-1* (lower panels) apices grown under SD conditions for the indicated times. DTB, days to bolting; DTF, days to the first flower opening; FUL, FRUITFULL; SD, short-day; TLN, total leaf number.
(TIF)

**S13 Fig. Flowering architecture of combinations between *spl15-1*, *ful-2*, and *mir172*.** Flowering architecture of indicated combinations between *spl15-1*, *ful-2*, and *mir172* in (A–C) SDs and (D–H) LDs. LD, long-day; SD, short-day.
(TIF)

**S1 Table. Results of statistical tests.**
(XLSX)

**S2 Table. Primers used in this study.**
(XLSX)

**S3 Table. Number of mapped reads for RNA-seq of Col-0 and *mir172abd* apices in LDs and SDs.** LD, long-day; RNA-seq, RNA-sequencing; SD, short-day.
(XLSX)

**S1 Data. Data underlying Fig 1A–1F, Fig 4D, Fig 5C and 5D, S9O Fig, and S11B and S11C Fig.**
(XLSX)

**S2 Data. Data underlying Fig 2B and 2D, Fig 3A and 3B, Fig 3D and 3E, Fig 5E and 5F, Fig 6A, Fig 6K and 6L, Fig 7A and 7B, S7D Fig, and S12A–S12E Fig.**
(XLSX)

**S3 Data. Data underlying S4F–S4L Fig.**
(TIF)

# Acknowledgments

We thank the staff of the Plant Cultivation Facilities at the MPIPZ and Kerstin Luxa for assistance with genotyping. We also thank the members of the Coupland group for helpful and engaging discussions during this project. We thank Jiawei Wang for helpful and interesting discussions of this manuscript and for sharing results prior to publication, and Kishore Panigrahi for discussions.

# Author Contributions

**Conceptualization:** Diarmuid S. Ó'Maoiléidigh, George Coupland.

**Data curation:** Diarmuid S. Ó'Maoiléidigh, Enric Bertran Garcia de Olalla.

**Formal analysis:** Diarmuid S. Ó'Maoiléidigh, Enric Bertran Garcia de Olalla, Alice Vayssières, George Coupland.

**Funding acquisition:** Diarmuid S. Ó'Maoiléidigh, George Coupland.

**Investigation:** Diarmuid S. Ó'Maoiléidigh, Annabel D. van Driel, Anamika Singh, Qing Sang, Nolwenn Le Bec, Coral Vincent, Enric Bertran Garcia de Olalla, Alice Vayssières, Maida Romera Branchat, Edouard Severing, Rafael Martinez Gallegos.

**Methodology:** Diarmuid S. Ó'Maoiléidigh, George Coupland.

**Project administration:** George Coupland.

**Resources:** Diarmuid S. Ó'Maoiléidigh, George Coupland.

**Supervision:** Diarmuid S. Ó'Maoiléidigh, Alice Vayssières, George Coupland.

**Visualization:** Diarmuid S. Ó'Maoiléidigh, Enric Bertran Garcia de Olalla.

**Writing – original draft:** Diarmuid S. Ó'Maoiléidigh, George Coupland.

**Writing – review & editing:** Diarmuid S. Ó'Maoiléidigh, Enric Bertran Garcia de Olalla, George Coupland.

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
