## [Editor Report · Decision Letter 0]

14 Jul 2020

Dear Dr Coupland, 

Thank you for submitting your manuscript entitled "Analysis of the MIR172 family defines transcriptional and post-transcriptional mechanisms that coordinately regulate APETALA2 to control floral transition of Arabidopsis" for consideration as a Research Article by PLOS Biology.

Your manuscript has now been evaluated by the PLOS Biology editorial staff as well as by an academic editor with relevant expertise and I am writing to let you know that we would like to send your submission out for external peer review.

Please re-submit your manuscript within two working days, i.e. by Jul 16 2020 11:59PM.

Kind regards,

Ines

--

Ines Alvarez-Garcia, PhD

Senior Editor

PLOS Biology

---

## [Decision Letter · Decision Letter 1]

23 Sep 2020

Dear Dr Coupland,

Thank you very much for submitting your manuscript "Analysis of the MIR172 family defines transcriptional and post-transcriptional mechanisms that coordinately regulate APETALA2 to control floral transition of Arabidopsis" for consideration as a Research Article at PLOS Biology. Thank you also for your patience as we completed our editorial process, and please accept my sincere apologies for the delay in providing you with our decision. Your manuscript has been evaluated by the PLOS Biology editors, an Academic Editor with relevant expertise, and by three independent reviewers.

The reviews are attached below. You will see that the reviewers find your results interesting and novel and think it is worth pursuing publication of the manuscript in PLOS Biology. Thus we are pleased to offer you the opportunity to address the points raised by the reviewers in a revised version that we anticipate should not take you very long. We will then assess your revised manuscript and your response to the reviewers' comments and we may consult the reviewers again.

We expect to receive your revised manuscript within 1 month.

**IMPORTANT - SUBMITTING YOUR REVISION**

*Resubmission Checklist*

*Published Peer Review*

*PLOS Data Policy*

*Blot and Gel Data Policy*

Sincerely,

Ines

--

Ines Alvarez-Garcia, PhD,

Senior Editor,

ialvarez-garcia@plos.org,

PLOS Biology

Reviewers’ comments

Rev. 1:

In this manuscript, the authors investigated spatiotemporal expressions and functions of MIR172 genes, thereby establishing genetic frame of transcriptional and post-transcriptional regulation of AP2 through FUL and miR172s during floral transition. Each MIRNA172-driven reporter analysis reveals that MIR172 genes are dynamically expressed in the shoot apex. Characterization of individual mutant allele of MIR172 genes generated by CRISPR-Cas9 technology and following genetic analysis show that each miRNA plays a redundant and/or specific role in floral induction in response to photoperiod. FUL and miR172s additively regulate floral induction, whose expressions are modulated by SPL15 in part. Together the authors propose that spatiotemporal regulation of AP2-like transcription factors critical for floral transition are coordinated by FUL and miR172s at the transcriptional and at the post-transcriptional level, respectively.

Majority of miRNAs in plants is composed of gene families but, to date, functional specificities of individual miRNAs has been poorly understood. This study clearly provides the functional diversity of MIR172 gene family, which eventually establish a genetic frame of regulatory robustness for flowering time. This is a well-written and organized paper. The genetic analysis is sophisticated and related genetic data including CRISPR-induced mutants of each miR172 are exciting and convincing. I have a few minor comments.

1. In Fig4, the analysis of the each MIR172 promoter activity suggests that MIR172B could be the main contributor for the miR172 pool, which was also supported by the measurement of mature miR172 levels in miR172b and miR172 quintuple mutant. How is the accumulation level of mature miR172 in other miR172 single mutants? Since the level of miRNA can be determined not only at the transcriptional level but also at the processing step, I would be nice to clarify it.

2. In Fig 6A, please determine the AP2 expression level (may include other AP2-like genes) in the shoot apex in ful, mir172abcd and ful mir172abcd in a similar way of Fig 1A-F. Although it is known that FUL and miR172 repress AP2-like genes, it still needs to show that the additively delayed flowering in ful mir172abcd mutants is indeed due to the mis-regulation of AP2-like genes in the shoot apex to support the author's conclusion.

3. Ref.33 is not the right reference. The authors probably mean this paper: POWERDRESS and Diversified Expression of the MIR172 Gene Family Bolster the Floral Stem Cell Network.

Rev. 2:

This paper represents a comprehensive analysis of the function of the miR172 genes in Arabidopsis and the role/integration into the flowering pathway. The authors produce mutations in all five MIR172 genes via CRISPR gene editing, and prepared quintuple mutants, a first for plant miRNA biology. They perform some lovely experiments with the NVG reporter fusion, and then carry out a detailed genetic analysis using ap2, ful and spl alleles and devise a model of how these genes are interacting to control flowering time. All the experiments to me look very well done, very nicely presented and well explained. This study makes a nice contribution to how genes are interacting to control flowering time in Arabidopsis and given all these factors appear strongly conserved this information will like be able to be used as a guide to understand flowering time in other plants. This study really clarifies the role of miR172 in flowering time and its role at the shoot apex.

I have no major comments about the science, I think this is a strong contribution worth publishing. My one trivial comment is in the first paragraph of the intro about feedback loops between miRNA and TF; this does not really appear to relate to plant biology, but rather animals, so seems a bit misleading.

Typo, missing sentence? - Furthermore,

Rev. 3:

Review of "Analysis of the MIR172 family defines transcriptional and posttranscriptional mechanisms that coordinately regulate APETALA2 to control floral transition of Arabidopsis" by Diarmuid S. Ó'Maoiléidigh et al.

This work analyzed the MIR172 family of small RNAs in Arabidopsis thaliana, which is comprised by five members. The authors used CRISPR-Cas9-induced mutations to inactivate each family member and prepared reporters to analyze the expression pattern of each MIRNA by confocal microscopy. The authors perform genetic analyses combining the mir172 mutants generated here with other known regulators such as the transcription factors FUL and SPL15 to dissect the participation of the pathway in the promotion of flowering.

The generation of mutants for all MIR172 family members and the thorough and laborious genetic analysis performed here are very well done and I think the results will be a of wide interest to the scientific community.

I have, however, some comments about the manuscript.

1) Figure 2E, I think there is something odd with the Total Leaf Number of transgenic plants overexpressing wild-type MIR172 precursors as they usually result in plants with much lower number of leaves. That wt MIR172 precursors have little effect, makes it difficult to discard a partial activity of the mir172 mutants. I think the authors should determine the mature miR172 levels in the CRISPR-Cas9-induced mutants to additionally validate the penetrance of the employed strategy.

2) Overall, I think it is important to improve the description of the expression of miR172-target genes in a mir172 mutant background. For example, can the levels of AP2-like transcription factors be determined by RT-qPCR in wt and mir172 mutants (using plants in the same developmental phase)?

In Figure 5, the pattern of AP2-Venus (protein) reporter (Fig 5A) does not match the pattern of AP2 RNA (Fig 5B), even in the mir172 mutant. For example, the AP2-Venus protein seems to be expressed in the meristem, while the AP2 RNA is present in other tissues much stronger than in the meristem. Furthermore, AP2-venus has a peak at 14d in wt, while it has a peak at 19d in mir172abd mutants. Wouldn't it be expected that the miR172-target AP2 accumulates at higher levels at earlier stages in the mir172 mutant compared to wt? Please revise and discuss these data.

3) Figure 4E. Please, indicate whether there are differences in the anatomy and/or developmental phase of the plants. This also applies to Figure 5, as differences in gene expression can be explained by differences in the plant developmental phase and/or anatomy rather than a consequence of miR172 activity.

Additional comments:

4) The reporters for MIRNA expression were designed by replacing the precursor sequence with NLS-Venus-GUS. I wonder whether this strategy could result in a reporter mRNA having long 5' UTR with spurious ATGs that will in turn affect its expression. Please comment.

5) Figures have low resolution. Figure 3C, the phenotype indicated by the arrow is not clear.

6) "Furthermore, The TLN and DTB of mir172ab", there is a typo ", The".

7) "…transition at the SAM, and initiated flowering from axillary branches, although most of these plants did eventually flower from the SAM as well (Fig 13A-C)." - there is a typo Fig S12A-C

8) "These results indicate that transcriptional activation of FUL and MIR172 by SPL15 may be essential for flowering of the SAM in some plants" -- typo FUL/MIR172 should be in italics

---

## [Editor Report · Decision Letter 2]

23 Nov 2020

Dear Dr Coupland,

Thank you for submitting your revised Research Article entitled "Analysis of the MIR172 family defines transcriptional and post-transcriptional mechanisms that coordinately regulate APETALA2 to control floral transition of Arabidopsis" for publication in PLOS Biology. I have obtained advice from the Academic Editor and discussed the revision with the team of editors. 

We're delighted to let you know that we're now editorially satisfied with your manuscript. However, we would like you to consider a change in the title to:

"The five MIR172 family members coordinately regulate APETALA2 to control floral transition of Arabidopsis"

Before we can formally accept your paper and consider it "in press", we also need to ensure that your article conforms to our guidelines. A member of our team will be in touch shortly with a set of requests. As we can't proceed until these requirements are met, your swift response will help prevent delays to publication. Please also make sure to address the data and other policy-related requests noted at the end of this email.

- a cover letter that should detail your responses to any editorial requests, if applicable

*Copyediting*

*Published Peer Review History*

*Early Version*

Sincerely,

Ines

--

Ines Alvarez-Garcia, PhD

Senior Editor,

PLOS Biology

Fig. 1A-F; Fig. 2B-D; Fig. 3A, B, D, E; Fig. 4D; Fig. 5C-F; Fig. 6A, E-L; Fig. 7A, B; Fig. S7C, D; Fig. S9O; Fig. S11B and Fig. S12A-E

Also, please make publicly available the data deposited in BioProject NCBI database (SubmissionID: SUB7773938, BioProject ID: PRJNA646473 and BioProject ID: PRJNA669254).

---

## [Editor Report · Decision Letter 3]

29 Dec 2020

Dear Dr. Coupland,

I am writing concerning your manuscript submitted to PLOS Biology, entitled “Systematic analyses of the MIR172 family members of Arabidopsis define their distinct roles in regulation of APETALA2 during floral transition.”

We have now completed our final technical checks and have approved your submission for publication. You will shortly receive a letter of formal acceptance from the editor.

Kind regards,

PLOS Biology